# Delving into Temperature Scaling for Adaptive Conformal Prediction

## Abstract

Conformal prediction, as an emerging uncertainty qualification technique, constructs prediction sets that are guaranteed to contain the true label with pre-defined probability. Previous works often employ temperature scaling to calibrate the classifier, assuming that confidence calibration can benefit conformal prediction. In this work, we empirically show that current confidence calibration methods (e.g., temperature scaling) normally lead to larger prediction sets in adaptive conformal prediction. Theoretically, we prove that a prediction with higher confidence could result in a smaller prediction set on expectation. Inspired by the analysis, we propose **Conformal Temperature Scaling** (ConfTS), a variant of temperature scaling that aims to improve the efficiency of adaptive conformal prediction. Specifically, ConfTS optimizes the temperature value by minimizing the gap between the threshold and the non-conformity score of the ground truth for a held-out validation dataset. In this way, the temperature value obtained would lead to an optimal set of high efficiency without violating the marginal coverage property. Extensive experiments demonstrate that our method can enhance adaptive conformal prediction methods. When averaged across six different architectures, ConfTS reduces the size of APS and RAPS on ImageNet by nearly 50% at an error rate of $\alpha = 0.1$.

## 1 Introduction

Ensuring the reliability of model predictions is crucial for the safe deployment of machine learning such as autonomous driving (Bojarski et al., 2016) and medical diagnostics (Caruana et al., 2015). Numerous methods have been developed to estimate uncertainty and incorporate it into predictive models, including confidence calibration (Guo et al., 2017) and Bayesian neural networks (Smith, 2013). However, these approaches lack theoretical guarantees of model performance. *Conformal prediction*, on the other hand, offers a systematic approach to construct prediction sets that contain ground-truth labels with a desired coverage guarantee (Vovk et al., 2005; Shafer & Vovk, 2008; Balasubramanian et al., 2014; Angelopoulos & Bates, 2021; Manokhin, 2022). This framework thus provides trustworthiness in real-world scenarios where wrong predictions are dangerous and costly.

In the literature, conformal prediction is frequently associated with *confidence calibration*, which expects the model to predict softmax probabilities that faithfully estimate the true correctness (Wang et al., 2021; Wei et al., 2022; Yuksekgonul et al., 2023; Wang, 2023; Wang et al., 2024). For example, existing conformal prediction methods usually employ temperature scaling (Guo et al., 2017), a post-hoc method that rescales the logits with a scalar temperature, to process the model output for a better calibration performance (Angelopoulos et al., 2021b; Lu et al., 2022; 2023; Gibbs et al., 2023). The underlying hypothesis is that well-calibrated models could yield precise probability estimates, thus enhancing the reliability of generated prediction sets. However, the rigorous impacts of current confidence calibration techniques on conformal prediction remain ambiguous in the literature, which motivates our analysis of the connection between conformal prediction and confidence calibration.

In this paper, we empirically show that existing methods of confidence calibration increase the size of prediction sets generated by adaptive conformal prediction methods. Moreover, high-confident predictions, rescaled by a small temperature value (Guo et al., 2017), often result in efficient prediction sets, while maintaining the desired coverage. To explain this phenomenon, we theoretically prove that a prediction, applied with a smaller temperature, could result in a more efficient prediction set on expectation. However, simply adopting an extremely small temperature value may result in invalid

and meaningless prediction sets since some tail probabilities would be truncated to zero due to the finite precision problem. Given these findings, our goal is to automatically search for a temperature value that can improve the efficiency of prediction sets for adaptive conformal prediction methods.

To this end, we propose a variant of temperature scaling, *Conformal Temperature Scaling* (ConfTS), which optimizes the temperature value by minimizing the efficiency gap, i.e., the deviation between the threshold and the non-conformity score of the ground truth. We calculate the efficiency gap with the non-randomized APS score (Romano et al., 2020) for a hold-out dataset. In effect, ConfTS optimizes the temperature value to improve the efficiency of prediction sets, preserving the marginal coverage. Notably, our method is compatible with the original temperature scaling designed for confidence calibration, as we can pick temperature values according to the purpose during inference.

Extensive experiments show that ConfTS can effectively enhance existing adaptive conformal prediction techniques. In particular, our method drastically improves the efficiency of the prediction sets for APS (Romano et al., 2020) and RAPS (Angelopoulos et al., 2021b). For instance, using ViT-B-16 (Dosovitskiy et al., 2021) on ImageNet (Deng et al., 2009), ConfTS reduces the average set size of APS at $\alpha = 0.1$ from 14.6 to 2.3, and declines that of RAPS from 6.9 to 1.8. Furthermore, ConfTS improves the conditional coverage of APS, and enhances the performance of the training-time method, ConfTr (Stutz et al., 2022). In practice, **our approach is straightforward to implement within deep learning frameworks, requiring no additional computational costs on temperature scaling**.

## 2 PRELIMINARY

In this work, we consider the multi-class classification task with $K$ classes. Let $\mathcal{X} \subset \mathbb{R}^d$ be the input space and $\mathcal{Y} := \{1, 2, \cdots, K\}$ be the label space. We represent a pre-trained classification model by $f : \mathcal{X} \to \mathbb{R}^K$. Let $(X, Y) \sim \mathcal{P}_{\mathcal{X}\mathcal{Y}}$ denote a random data pair sampled from a joint data distribution $\mathcal{P}_{\mathcal{X}\mathcal{Y}}$, and $\boldsymbol{f}_y(\boldsymbol{x})$ denote the $y$-th element of logits vector $\boldsymbol{f}(\boldsymbol{x})$ with a instance $\boldsymbol{x}$. Normally, the conditional probability of class $y$ is approximated by the softmax probability output $\boldsymbol{\pi}(\boldsymbol{x})$ defined as:

$$\mathbb{P}\{Y = y | X = x\} \approx \pi_y(\boldsymbol{x}; t) = \sigma(f(\boldsymbol{x}); t)_y = \frac{e^{f_y(\boldsymbol{x})/t}}{\sum_{i=1}^{K} e^{f_i(\boldsymbol{x})/t}}, \quad (1)$$

where $\sigma$ is the softmax function and $t$ denotes the temperature parameter (Guo et al., 2017). The temperature softens the output probability with $t > 1$ and sharpens the probability with $t < 1$. After training the model, the temperature can be tuned on a held-out validation set by optimization methods.

**Conformal prediction.** To provide theoretical guarantees for model predictions, conformal prediction (Vovk et al., 2005) is designated for producing prediction sets that contain ground-truth labels with a desired probability rather than predicting one-hot labels. In particular, the goal of conformal prediction is to construct a set-valued mapping $\mathcal{C} : \mathcal{X} \to 2^{\mathcal{Y}}$ that satisfies the *marginal coverage*:

$$\mathbb{P}(Y \in \mathcal{C}(\boldsymbol{x})) \geq 1 - \alpha, \quad (2)$$

where $\alpha \in (0, 1)$ denotes a user-specified error rate, and $\mathcal{C}(\boldsymbol{x}) \subset \mathcal{Y}$ is the generated prediction set.

Before deployment, conformal prediction begins with a calibration step, using a held-out calibration set $\mathcal{D}_{cal} := \{(\boldsymbol{x}_i, y_i)\}_{i=1}^{n}$. We calculate the non-conformity score $s_i = \mathcal{S}(\boldsymbol{x}_i, y_i)$ for each example $(\boldsymbol{x}_i, y_i)$, where $s_i$ is a measure of deviation between an example and the training data, which we will specify later. Then, we determine the $1 - \alpha$ quantile of the non-conformity scores as a threshold:

$$\tau = \inf \left\{ s : \frac{|\{i : \mathcal{S}(\boldsymbol{x}_i, y_i) \leq s\}|}{n} \geq \frac{\lceil (n+1)(1-\alpha) \rceil}{n} \right\}. \quad (3)$$

For a test instance $\boldsymbol{x}_{n+1}$, we first calculate the non-conformity score for each label in $\mathcal{Y}$, and then construct the prediction set $\mathcal{C}(\boldsymbol{x}_{n+1})$ by including labels whose non-conformity score falls within $\tau$:

$$\mathcal{C}(\boldsymbol{x}_{n+1}) = \{y \in \mathcal{Y} : \mathcal{S}(\boldsymbol{x}_{n+1}, y) \leq \tau\}. \quad (4)$$

In this paper, we focus on *adaptive* conformal prediction methods, which are designed to guarantee conditional coverage by improving adaptiveness (Romano et al., 2020). However, they usually suffer from inefficiency in practice: these methods commonly produce large prediction sets (Angelopoulos

Table 1: The performance of APS and RAPS on CIFAR-100 and ImageNet dataset using various post-hoc calibration methods. In particular, we apply vector scaling (VS), Platt scaling (PS), and temperature scaling (TS). We do not employ calibration techniques in the baseline (Base). We repeat each experiment for 20 times. "↓" indicates smaller values are better. "▲" and "▼" indicate whether the performance is superior/inferior to the baseline. **Bold** numbers are superior results. The results show that all post-hoc confidence calibration methods deteriorate the efficiency of APS and RAPS.

| Datasets | Metrics | | ResNet18 | | | | ResNet50 | | | | ResNet101 | | | |
|---|---|---|---|---|---|---|---|---|---|---|---|---|---|---|
| | | | Base | VS | PS | TS | Base | VS | PS | TS | Base | VS | PS | TS |
| CIFAR-100 | | Accuracy | 0.76 | 0.75 | 0.76 | 0.76 | 0.77 | 0.77 | 0.77 | 0.77 | 0.78 | 0.79 | 0.78 | 0.78 |
| | | ECE(%)↓ | 5.68 | 3.67▲ | 4.20▲ | 4.29▲ | 8.79 | 3.62▲ | 3.81▲ | 4.06▲ | 10.8 | 3.27▲ | 3.83▲ | 3.62▲ |
| | APS | Coverage | 0.90 | 0.90 | 0.90 | 0.90 | 0.90 | 0.90 | 0.90 | 0.90 | 0.90 | 0.90 | 0.90 | 0.90 |
| | | Average size↓ | **8.73** | 8.94▼ | 10.1▼ | 10.0▼ | **4.91** | 6.69▼ | 7.75▼ | 7.35▼ | **4.01** | 5.77▼ | 6.99▼ | 6.66▼ |
| | RAPS | Coverage | 0.90 | 0.90 | 0.90 | 0.90 | 0.90 | 0.90 | 0.90 | 0.90 | 0.90 | 0.90 | 0.90 | 0.90 |
| | | Average size↓ | **4.14** | 4.40▼ | 4.71▼ | 4.67▼ | **2.63** | 3.58▼ | 3.72▼ | 3.85▼ | **2.27** | 3.39▼ | 3.54▼ | 3.53▼ |
| ImageNet | | Accuracy | 0.69 | 0.68 | 0.69 | 0.69 | 0.76 | 0.75 | 0.76 | 0.76 | 0.77 | 0.76 | 0.77 | 0.77 |
| | | ECE(%)↓ | 2.63 | 2.14▲ | 2.10▲ | 2.27▲ | 3.69 | 1.50▲ | 2.24▲ | 2.35▲ | 5.08 | 1.38▲ | 2.02▲ | 2.20▲ |
| | APS | Coverage | 0.90 | 0.90 | 0.90 | 0.90 | 0.90 | 0.90 | 0.90 | 0.90 | 0.90 | 0.90 | 0.90 | 0.90 |
| | | Average size↓ | **14.1** | 17.3▼ | 15.9▼ | 16.0▼ | **9.06** | 12.0▼ | 12.0▼ | 12.1▼ | **6.95** | 11.1▼ | 10.7▼ | 10.6▼ |
| | RAPS | Coverage | 0.90 | 0.90 | 0.90 | 0.90 | 0.90 | 0.90 | 0.90 | 0.90 | 0.90 | 0.90 | 0.90 | 0.90 |
| | | Average size↓ | **9.61** | 10.6▼ | 11.75▼ | 11.30▼ | **5.99** | 7.56▼ | 7.52▼ | 7.16▼ | **4.82** | 6.86▼ | 6.85▼ | 6.59▼ |

et al., 2021b). In particular, we take the two representative methods: *Adaptive Prediction Sets (APS)* (Romano et al., 2020) and *Regularized Adaptive Prediction Sets (RAPS)* (Angelopoulos et al., 2021b).

**Adaptive Prediction Set (APS).** (**Romano et al., 2020**) In the APS method, the non-conformity score of a data pair $(\boldsymbol{x}, y)$ is calculated by accumulating the sorted softmax probability, defined as:

$$\mathcal{S}_{APS}(\boldsymbol{x}, y) = \pi_{(1)}(\boldsymbol{x}) + \cdots + u \cdot \pi_{o(y, \pi(\boldsymbol{x}))}(\boldsymbol{x}), \tag{5}$$

where $\pi_{(1)}(\boldsymbol{x}), \pi_{(2)}(\boldsymbol{x}), \cdots, \pi_{(K)}(\boldsymbol{x})$ are the sorted softmax probabilities in descending order, and $o(y, \pi(\boldsymbol{x}))$ denotes the order of $\pi_y(\boldsymbol{x})$, i.e., the softmax probability for the ground-truth label $y$. In addition, the term $u$ is an independent random variable that follows a uniform distribution on $[0, 1]$.

**Regularized Adaptive Prediction Set (RAPS).** (**Angelopoulos et al., 2021b**) The non-conformity score function of RAPS encourages a small set size by adding a penalty, as formally defined below:

$$\mathcal{S}_{RAPS}(\boldsymbol{x}, y) = \pi_{(1)}(\boldsymbol{x}) + \cdots + u \cdot \pi_{o(y, \pi(\boldsymbol{x}))}(\boldsymbol{x}) + \lambda \cdot (o(y, \pi(\boldsymbol{x})) - k_{reg})^+, \tag{6}$$

where $(z)^+ = \max\{0, z\}$, $k_{reg}$ controls the number of penalized classes, and $\lambda$ is the penalty term.

Notably, both methods incorporate a uniform random variable $u$ to achieve exact $1 - \alpha$ coverage (Angelopoulos et al., 2021b). Moreover, we use *coverage*, *average size*, and *size-stratified coverage violation (SSCV)* (Angelopoulos et al., 2021b) to assess the marginal and conditional coverage, as well as the efficiency of prediction sets. A detailed description of the metrics is provided in Appendix A.

## 3 MOTIVATION

### 3.1 ADAPTIVE CONFORMAL PREDICTION WITH CALIBRATED PREDICTION

*Confidence calibration* (Guo et al., 2017) expects the model to predict softmax probabilities that faithfully estimate the true correctness: $\forall p \in [0, 1], \mathbb{P}\{Y = y | \pi_y(\boldsymbol{x}) = p\} = p$. To measure the miscalibration, *Expected Calibration Error* (ECE) (Naeini et al., 2015) averages the difference between the accuracy $\mathrm{acc}(\cdot)$ and confidence $\mathrm{conf}(\cdot)$ in $M$ bins: $\mathrm{ECE} = \sum_{m=1}^{M} \frac{|B_m|}{|\mathcal{I}_{test}|} |acc(B_m) - conf(B_m)|$, where $B_m$ denotes the $m$-th bin. In conformal prediction, previous work claims that deep learning models are often badly miscalibrated, leading to large prediction sets that do not faithfully articulate the uncertainty of the model (Angelopoulos et al., 2021b). To address the issue, researchers usually employ temperature scaling (Guo et al., 2017) to process the model outputs for better calibration performance. However, the precise impacts of current confidence calibration techniques on adaptive conformal prediction remain unexplored, which motivates our investigation into this connection.

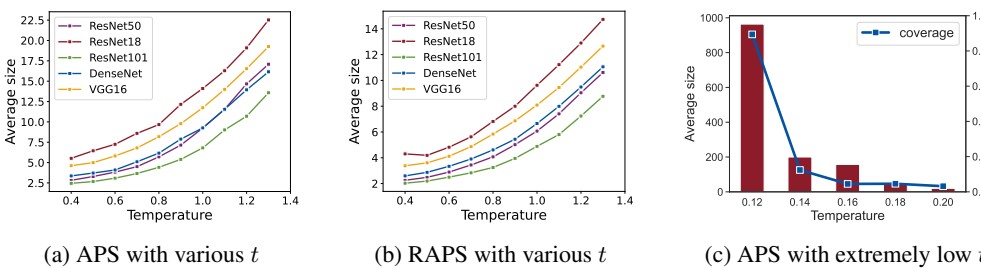

(a) APS with various $t$      (b) RAPS with various $t$      (c) APS with extremely low $t$

Figure 1: (a) & (b): The performance of APS and RAPS with different temperatures on ImageNet. The results show that high-confidence predictions, with a small temperature, lead to efficient prediction sets. (c): The performance of APS for ResNet18 on ImageNet with *extremely* low temperatures. In this setting, APS generates large prediction sets with conservative coverage due to finite precision.

**Confidence calibration methods deteriorate the efficiency of prediction sets.** To figure out the correlation between confidence calibration and adaptive conformal prediction, we incorporate various confidence calibration methods to adaptive conformal predictors for classification models on CIFAR-100 (Krizhevsky et al., 2009) and ImageNet (Deng et al., 2009). Specifically, we use 6 calibration methods, including 4 post-hoc methods – *vector scaling* (Guo et al., 2017), *Platt scaling* (Platt et al., 1999), *temperature scaling* (Guo et al., 2017), *Bayesian methods* (Daxberger et al., 2021), and 2 training methods – *label smoothing* (Szegedy et al., 2016), *mixup* (Zhang et al., 2018). More details of calibration methods and setups are presented in Appendix B and Appendix C, respectively.

In Table 1, we present the performance of calibration and conformal prediction using APS and RAPS with various post-hoc calibration methods. The results show that the influences of those calibration methods are consistent: **models calibrated by these techniques generate large prediction sets** with lower ECE (i.e., better calibration). For example, on CIFAR-100 with ResNet50, Platt scaling enlarges the average size of prediction sets of APS from 4.91 to 7.75, while decreasing the ECE from 8.79% to 3.81%. In addition, incorporating calibration methods into conformal prediction does not violate the $1 - \alpha$ marginal coverage as the assumption of data exchangeability is still satisfied: we use a hold-out validation dataset for conducting confidence calibration methods. The same conclusion can be obtained for training-time and Bayesian-based calibration methods, as shown in Appendix D.

Overall, we empirically show that current confidence calibration methods negatively impact the efficiency of prediction sets, challenging the conventional practice of employing temperature scaling in adaptive conformal prediction. While confidence calibration methods are primarily designed to address overconfidence, we conjecture that high confidence may enhance prediction sets in efficiency.

### 3.2 Adaptive Conformal prediction with high-confidence prediction

In this section, we investigate how the high-confidence prediction influences the adaptive conformal prediction. In particular, we employ temperature scaling with different temperatures $t \in [0.4, 0.5, \cdots, 1.3]$ (defined in Eq. (1)) to control the confidence level. The analysis is conducted on the ImageNet dataset with various model architectures, using APS and RAPS at $\alpha = 0.1$.

**High confidence enhances the efficiency of adaptive conformal prediction.** In Figures 1a and 1b, we present the average size of prediction sets generated by APS and RAPS under various temperature values $t$. The results show that a highly-confident model, produced by a small temperature value, would decrease the average size of prediction sets. For example, using VGG16, the average size is reduced by four times – from 20 to 5, with the decrease of the temperature value from 1.3 to 0.5. There naturally arises a question: *is it always better for efficiency to take smaller temperature values?*

In Figure 1c, we report the average size of prediction sets produced by APS on ImageNet with ResNet18, using *extremely* small temperatures (i.e. $t \in \{0.12, 0.14, \cdots, 0.2\}$). Different from the above, APS generates larger prediction sets with smaller temperatures in this range, even leading to conservative coverage. This problem stems from floating point numerical errors caused by finite precision (see Appendix F for a detailed explanation). The phenomenon indicates that it is non-trivial to find the optimal temperature value for the highest efficiency of adaptive conformal prediction.

### 3.3 THEORETICAL EXPLANATION

Intuitively, confident predictions are expected to yield smaller prediction sets than conservative ones. Here, we provide a theoretical justification for this by showing how the reduction of temperature decreases the average size of prediction sets in the case of non-randomized APS (simply omit the random term in Eq. (5)). We start by analyzing the relationship between the temperature $t$ and the APS score. For simplicity, assuming the logits vector $\boldsymbol{f}(\boldsymbol{x}) := [f_1(\boldsymbol{x}), f_2(\boldsymbol{x}), \ldots, f_K(\boldsymbol{x})]^T$ satisfies $f_1(\boldsymbol{x}) > f_2(\boldsymbol{x}) > \cdots > f_K(\boldsymbol{x})$, then, the non-randomized APS score for class $k \in \mathcal{Y}$ is given by:

$$\mathcal{S}(\boldsymbol{x}, k, t) = \sum_{i=1}^{k} \frac{e^{f_i(\boldsymbol{x})/t}}{\sum_{j=1}^{K} e^{f_j(\boldsymbol{x})/t}}. \tag{7}$$

Then, we can derive the following proposition on the connection of the temperature and the score:

**Proposition 3.1.** *For instance $\boldsymbol{x} \in \mathcal{X}$, let $\mathcal{S}(\boldsymbol{x}, k, t)$ be the non-conformity score function of an arbitrary class $k \in \mathcal{Y}$, defined as in Eq. 7. Then, for any temperature $t_0 \in \mathbb{R}^+$ and $\forall t \in (0, t_0)$, we have*

$$\mathcal{S}(\boldsymbol{x}, k, t_0) \leq \mathcal{S}(\boldsymbol{x}, k, t).$$

The proof is provided in Appendix G.1. In Proposition 3.1, we show that the APS score increases as temperature decreases, and vice versa. Then, we fix a temperature $t_0 \in \mathbb{R}^+$, and further define $\epsilon(k, t) = \mathcal{S}(\boldsymbol{x}, k, t) - \mathcal{S}(\boldsymbol{x}, k, t_0) \geq 0$ as the difference of the APS scores. As a corollary of Proposition 3.1, we conclude that $\epsilon(k, t)$ is negatively correlated with the temperature $t$. We provide the proof for this corollary in Appendix G.2. The corollary is formally stated as follows:

**Corollary 3.2.** *For any sample $\boldsymbol{x} \in \mathcal{X}$ and a fixed temperature $t_0$, the difference $\epsilon(k, t)$ is a decreasing function with respect to $t \in (0, t_0)$.*

In the following, we further explore how the change in the APS score affects the average size of the prediction set. In the theorem, we make two continuity assumptions on the CDF of the non-conformity score (see Appendix G.3), following prior works (Lei, 2014; Sadinle et al., 2019). Given these assumptions, we can derive an upper bound for the expected size of $\mathcal{C}(\boldsymbol{x}, t)$ for any $t \in (0, t_0)$:

**Theorem 3.3.** *Under assumptions in Appendix G.3, there exists constants $c_1, \gamma \in (0, 1]$ (defined in the above assumptions) such that*

$$\mathbb{E}_{\boldsymbol{x} \in \mathcal{X}}[|\mathcal{C}(\boldsymbol{x}, t)|] \leq K - \sum_{k \in \mathcal{Y}} c_1 [2\epsilon(k, t)]^{\gamma}, \quad \forall t \in (0, t_0).$$

**Interpretation.** The proof of Theorem 3.3 is presented in Appendix G.3. Through Theorem 3.3, we show that for any temperature $t$, the expected size of the prediction set $\mathcal{C}(\boldsymbol{x}, t)$ has an upper bound with respect to the non-conformity score deviation $\epsilon$. Recalling that $\epsilon$ increases with the decrease of temperature $t$, we conclude that a lower temperature $t$ results in a larger difference $\epsilon$, thereby narrowing the prediction set $\mathcal{C}(\boldsymbol{x}, t)$. Overall, the analysis shows that **high-confidence predictions, produced by a small temperature, could lead to efficient prediction sets on expectation**. Given the theoretical analysis, we propose to enhance the efficiency of adaptive conformal prediction by tuning the temperature. We proceed by introducing our method – *Conformal Temperature Scaling*.

## 4 METHOD

In the previous analysis, we show that temperature scaling optimized by negative log-likelihood deteriorates the efficiency of adaptive conformal prediction, while a small temperature can improve. Nevertheless, results in Figure 1c demonstrate that finding the optimal temperature for the highest efficiency is a non-trivial task. In this work, we propose *Conformal Temperature Scaling*, a variant of temperature scaling, to select an appropriate temperature for enhancing adaptive conformal prediction.

For a test example $(\boldsymbol{x}, y)$, conformal prediction aims to construct an *efficient* prediction set $\mathcal{C}(\boldsymbol{x})$ that contains the true label $y$. Thus, the *optimal prediction set* meeting this requirement is defined as:

$$\mathcal{C}^*(\boldsymbol{x}) = \{k \in \mathcal{Y} : \mathcal{S}(\boldsymbol{x}, k) \leq \mathcal{S}(\boldsymbol{x}, y)\}.$$

Specifically, the optimal prediction set is the smallest set that allows the inclusion of the ground-truth label. The concept of optimal prediction set naturally leads to a way for quantifying the redundancy

of generated prediction set $\mathcal{C}(\boldsymbol{x})$: we can compute the deviation between the size of the prediction set and that of the optimal prediction set $|\mathcal{C}(\boldsymbol{x})| - |\mathcal{C}^*(\boldsymbol{x})|$. However, it is challenging to perform optimization with the size difference due to its discrete property. To circumvent the issue, we convert the optimization objective of our method into a continuous loss function that is end-to-end trainable.

**Efficiency gap.** Recall that the prediction set is established through the $\tau$ calculated from the calibration set (Eq. (3)), the optimal set can be attained if the threshold $\tau$ well approximates the non-conformity score of the ground-truth label $\mathcal{S}(\boldsymbol{x}, y)$. Therefore, we can also measure the redundancy of the prediction set by the differences between thresholds $\tau$ and the score of true labels, defined as:

**Definition 4.1** (Efficiency Gap). *For an example* $(\boldsymbol{x}, y)$*, a threshold* $\tau$ *and a non-conformity score function* $\mathcal{S}(\cdot)$*, the efficiency gap of the instance* $\boldsymbol{x}$ *is given by:*

$$\mathcal{G}(\boldsymbol{x}, y, \tau) = \tau - \mathcal{S}(\boldsymbol{x}, y).$$

In particular, a positive efficiency gap indicates that the ground-truth label $y$ is included in the prediction set $y \in \mathcal{C}(\boldsymbol{x})$, and vice versa. To optimize for the optimal prediction set, we expect to increase the efficiency gap for samples with negative gaps and decrease it for those with positive gaps. We propose to accomplish the optimization by tuning the temperature $t$. This allows us to optimize the efficiency gap since $\mathcal{S}(\boldsymbol{x}, y)$ and $\tau$ are functions with respect to the temperature $t$ (see Eq. (7)).

**Conformal Temperature Scaling.** To this end, we propose our method – Conformal Temperature Scaling (dubbed **ConfTS**), which rectifies the objective function of temperature scaling through the efficiency gap. In particular, the loss function for ConfTS is formally given as follows:

$$\mathcal{L}_{\mathrm{ConfTS}}(\boldsymbol{x}, y; t) = (\tau(t) - \mathcal{S}(\boldsymbol{x}, y, t))^2, \tag{8}$$

where $\tau(t)$ is the conformal threshold and $\mathcal{S}(\boldsymbol{x}, y, t)$ denotes the *non-randomized* APS score of the example $(\boldsymbol{x}, y)$ with respect to $t$ (see Eq. (7)). By minimizing the mean squared error, the ConfTS loss encourages smaller prediction sets for samples with positive efficiency gaps, and vice versa.

**The optimization of ConfTS.** To preserve the exchangeability assumption, we tune the temperature to minimize the ConfTS loss on a held-out validation set. Following previous work (Stutz et al., 2022), we split the validation set into two subsets: one to compute $\tau(t)$, and the other to calculate the ConfTS loss with the obtained $\tau(t)$. Specifically, the optimization problem can be formulated as:

$$t^* = \arg\min_{t \in \mathbb{R}^+} \frac{1}{|\mathcal{D}_{\mathrm{loss}}|} \sum_{(\boldsymbol{x}_i, y_i) \in \mathcal{D}_{\mathrm{loss}}} \mathcal{L}_{\mathrm{ConfTS}}(\boldsymbol{x}_i, y_i; t), \tag{9}$$

where $\mathcal{D}_{\mathrm{loss}}$ denotes the subset for computing ConfTS loss. Trained with the ConfTS loss, we can optimize the temperature $t$ for adaptive prediction sets with high efficiency without violating coverage. Since the threshold $\tau$ is determined by the pre-defined $\alpha$, our ConfTS method can yield different temperature values for each $\alpha$. Notably, ConfTS offers compelling advantages as a post-hoc method:

- **Algorithm-agnostic**: ConfTS trained with APS score can improve other adaptive conformal prediction methods (e.g., RAPS) and is compatible with training-time methods such as ConfTr (Stutz et al., 2022) for improved efficiency. This is supported by Table 2 and Table 3.

- **Easy-to-use**: Our method enhances the efficiency of conformal prediction in a parameter-efficient fashion. This stands in contrast to training methods (Stutz et al., 2022), which require optimizing the full parameters of networks and may degrade the accuracy. Moreover, our ConfTS is free from hyper-parameter tuning, and requires low computational resources.

- **Flexible**: Our method does not conflict with confidence calibration, as it only replaces the temperature value. During inference, one may use different temperature values according to the objective, whether for improved calibration performance or efficient prediction sets.

## 5 EXPERIMENTS

In this section, we first verify the effectiveness of ConfTS in both post-hoc and training conformal prediction methods across several benchmark datasets. Then, we investigate the adaptivity property of prediction sets employed with ConfTS, focusing on the SSCV performance and adaptiveness. In addition, we present ablation studies examining the impact of validation and conformal set sizes, as well as the effect of using different non-conformity scores to compute the efficiency gap in ConfTS.

Table 2: Performance of ConfTS using APS and RAPS on ImageNet dataset. We repeat each experiment for 20 times. "*" denotes significant improvement (two-sample t-test at a 0.1 confidence level). "↓" indicates smaller values are better. **Bold** numbers are superior results. Results show that our ConfTS can improve the performance of APS and RAPS, maintaining the desired coverage rate.

| Model | Score | $\alpha = 0.1$ | | $\alpha = 0.05$ | |
| | | Coverage | Average size↓ | Coverage | Average size↓ |
| | | Base / ConfTS | | | |
| ResNet18 | APS | 0.900 / 0.900 | 14.09 / **7.531**\* | 0.951 / 0.952 | 29.58 / **19.59**\* |
| | RAPS | 0.900 / 0.900 | 9.605 / **5.003**\* | 0 .950 / 0.950 | 14.72 / **11.08**\* |
| ResNet50 | APS | 0.899 / 0.900 | 9.062 / **4.791**\* | 0.950 / 0.951 | 20.03 / **12.22**\* |
| | RAPS | 0.899 / 0.900 | 5.992 / **3.561**\* | 0.950 / 0.951 | 9.423 / **5.517**\* |
| ResNet101 | APS | 0.900 / 0.899 | 6.947 / **4.328**\* | 0.950 / 0.950 | 15.73 / **10.51**\* |
| | RAPS | 0.900 / 0.899 | 4.819 / **3.289**\* | 0.950 / 0.950 | 7.523 / **5.091**\* |
| DenseNet121 | APS | 0.900 / 0.899 | 9.271 / **4.746**\* | 0.950 / 0.949 | 20.37 / **11.47**\* |
| | RAPS | 0.900 / 0.900 | 6.602 / **3.667**\* | 0.949 / 0.949 | 10.39 / **6.203**\* |
| VGG16 | APS | 0.901 / 0.901 | 11.73 / **6.057**\* | 0.951 / 0.951 | 23.71 / **14.78**\* |
| | RAPS | 0.901 / 0.900 | 8.118 / **4.314**\* | 0.950 / 0.950 | 12.27 / **8.350**\* |
| ViT-B-16 | APS | 0.900 / 0.901 | 14.64 / **2.315**\* | 0.951 / 0.950 | 36.72 / **9.050**\* |
| | RAPS | 0.902 / 0.901 | 6.889 / **1.800**\* | 0.950 / 0.950 | 12.63 / **3.281**\* |
| *Average* | APS | 0.900 / 0.900 | 10.96 / **4.961**\* | 0.950 / 0.950 | 24.36 / **12.94**\* |
| | RAPS | 0.900 / 0.900 | 7.000 / **3.606**\* | 0.950 / 0.950 | 11.16 / **6.587**\* |

## 5.1 EXPERIMENTAL SETUP

**Datasets.** In this work, we verify the effectiveness of ConfTS on CIFAR-100 (Krizhevsky et al., 2009), ImageNet (Deng et al., 2009), and ImageNet-V2 (Recht et al., 2019). On ImageNet, we split the test dataset, including 50,000 images, into 10,000 images for the calibration set and 40,000 images for the test set. On CIFAR-100 and ImageNet-V2, we split the test dataset, including 10,000 figures, into 4,000 figures for the calibration set and 6,000 for the test set. Additionally, we split the calibration set into two subsets of equal size: one subset is the validation set to optimize the temperature value with ConfTS, while the other half is the conformal set for conformal calibration.

**Models.** For ImageNet and ImageNet-V2, we employ 6 pre-trained classifiers from TorchVision (Paszke et al., 2019) – ResNet18, ResNet50, ResNet101 (He et al., 2016), DenseNet121 (Huang et al., 2017), VGG16 (Simonyan & Zisserman, 2015) and ViT-B-16 (Dosovitskiy et al., 2021). We also utilize the same model architectures for CIFAR-100 and train them from scratch. The models are trained for 100 epochs using SGD with a momentum of 0.9, a weight decay of 0.0005, and a batch size of 128. We set the initial learning rate as 0.1 and reduce it by a factor of 5 at 60 epochs. For conformal training, we set the smoothing parameter $T = 0.1$, penalty term $\kappa = 1$, and hyperparameter $\lambda = 1$, using the same training setups. We conduct all the experiments on NVIDIA GeForce RTX 4090.

**Conformal prediction algorithms.** We leverage three adaptive conformal prediction methods: APS (Romano et al., 2020) and RAPS (Angelopoulos et al., 2021b) to generate prediction sets at error rate $\alpha \in \{0.1, 0.05\}$. In addition, we set the regularization hyper-parameter for RAPS to be: $k_{reg} = 1$ and $\lambda \in \{0.001, 0.002, 0.004, 0.006, 0.01, 0.015, 0.02\}$. For the evaluation metrics, we employ *coverage*, *average size*, and *SSCV* (see Appendix A) to assess the performance of prediction sets. All experiments are repeated 20 times with different seeds, and we report average performances.

## 5.2 MAIN RESULTS

**ConfTS improves current adaptive conformal prediction methods.** In Table 2, we present the performance of APS and RAPS ($\lambda = 0.001$) with ConfTS on the ImageNet dataset. A salient observation is that ConfTS drastically improves the efficiency of adaptive conformal prediction, while maintaining the marginal coverage. For example, on the ViT model at $\alpha = 0.05$, ConfTS reduces the average size of APS by 7 times - from 36.72 to 5.759. Averaged across six models, ConfTS improves the efficiency of APS by 58.3% at $\alpha = 0.1$. We observe similar results on CIFAR-100 and ImageNet-V2 dataset in Appendix H and Appendix I. Moreover, our ConfTS remains effective for RAPS

Table 3: The performance of ConfTS when applied to models trained with ConfTr loss on CIFAR-100. We repeat each experiment for 20 times. "*" denotes significant improvement (two-sample t-test at a 0.1 confidence level). **Bold** numbers are superior results. "↓" indicates smaller values are better. The results show that ConfTS boosts ConfTr: it generates efficient prediction sets for APS and RAPS.

| Model | Score | $\alpha = 0.1$ | | $\alpha = 0.05$ | |
|---|---|---|---|---|---|
| | | Coverage | Average size ↓ | Coverage | Average size ↓ |
| | | Baseline / ConfTS | | | |
| ResNet18 | APS | 0.899 / 0.899 | 6.670 / **5.827*** | 0.949 / 0.949 | 11.49 / **10.35*** |
| | RAPS | 0.900 / 0.900 | 5.848 / **4.799*** | 0.951 / 0.951 | 8.431 / **8.189*** |
| ResNet50 | APS | 0.900 / 0.901 | 5.754 / **5.075*** | 0.951 / 0.951 | 9.911 / **9.150*** |
| | RAPS | 0.900 / 0.900 | 5.186 / **4.324*** | 0.951 / 0.949 | 7.868 / **7.507*** |

across various penalty terms on ImageNet as shown in Appendix J. Furthermore, in Appendix K, we demonstrate that ConfTS can lead to small prediction sets for *SAPS* (Huang et al., 2024) – another adaptive conformal prediction technique. Overall, empirical results show that ConfTS consistently improves the efficiency of existing adaptive conformal prediction methods across various networks.

**ConfTS boosts training-time conformal prediction method.** Previous work (Stutz et al., 2022) proposes *Conformal Training (ConfTr)* which enhances the efficiency of prediction sets during training process. In this part, we investigate how our ConfTS interacts with ConfTr. In particular, we leverage ResNet18 and ResNet50 trained with ConfTr loss for 100 epochs on CIFAR-100, generating prediction sets with APS and RAPS ($\lambda = 0.001$) method at error rate $\alpha \in \{0.1, 0.05\}$. In Table 3, we present the performance of ConfTS when applied to networks trained with ConfTr loss. The results show that our ConfTS can boost the performance of ConfTr. For instance, on the ResNet50 model trained with ConfTr loss, at an error rate $\alpha = 0.1$, ConfTS reduces the average size of RAPS from 5.186 to 4.324. In summary, our findings highlight that ConfTS can effectively improve both post-hoc and training methods of conformal prediction.

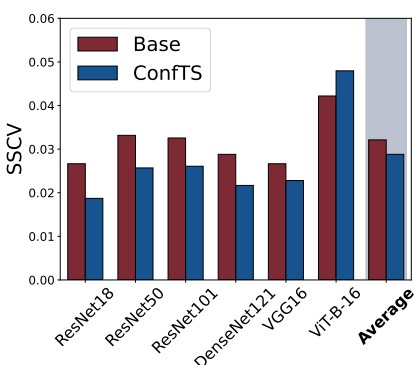

Figure 2: The SSCV performance of ConfTS using APS on ImageNet dataset. A smaller SSCV is better. The results show that ConfTS can improve the conditional coverage performance of APS.

**ConfTS enhances the conditional coverage.** The *conditional coverage* (Vovk, 2012) requires conformal prediction methods to satisfy the marginal coverage at instance level. The *size-stratified coverage violation (SSCV)* (Angelopoulos et al., 2021b) is often employed to evaluate the conditional coverage of prediction sets:

$$\text{SSCV} = \sup_j |\frac{|\{i \in S_j : y_i \in \mathcal{C}(\boldsymbol{x}_i)\}|}{|S_j|} - (1 - \alpha)|,$$

where $\{S_i\}_{i=1}^{N_s}$ is a disjoint set-size strata, satisfying $\bigcup_{i=1}^{N_s} S_i = \{1, 2, \cdots, |\mathcal{Y}|\}$. Specifically, a lower SSCV value indicates better conditional coverage performance. Following prior work (Angelopoulos et al., 2021b), we set the partitioning of the set sizes as: 0-1, 2-3, 4-10, 11-100, and 101-1000. Figure 2 presents the SSCV performance of ConfTS using APS on ImageNet at $\alpha = 0.05$. The results show that ConfTS can enhance conditional coverage in most cases. For example, on ResNet50, ConfTS reduces the SSCV of APS from 0.033 to 0.025. Moreover, in Appendix M, we provide a synthetic experiment that shows that ConfTS is particularly effective in improving conditional coverage when the classification task is more deterministic. Overall, these findings highlight that ConfTS improves both conditional coverage and the efficiency of adaptive conformal prediction.

**ConfTS maintains the adaptiveness.** Adaptiveness (Romano et al., 2020; Angelopoulos et al., 2021b; Seedat et al., 2023) requires prediction sets to communicate instance-wise uncertainty: easy

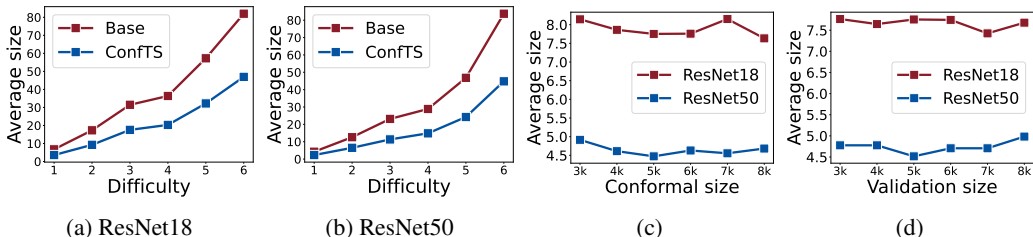

(a) ResNet18       (b) ResNet50       (c)       (d)

Figure 3: (a)&(b): Average sizes of examples with different difficulties using APS on ResNet18 and ResNet50 respectively. Results show that ConfTS can maintain adaptiveness. (b)&(c) Average sizes of APS employed with ConfTS under various sizes of (b) conformal dataset (c) validation dataset. Results show that our ConfTS is robust to variations in the validation and conformal dataset size.

Table 4: The performance of ConfTS using various non-conformity scores to compute the efficiency gap. We consider standard APS and RAPS score as well as their non-randomized variants. Each experiment is repeated 20 times. "Avg.size" and "Cov." represent the results of average size and coverage, and 'Base' presents the results without ConfTS. The non-conformity scores in rows indicate the methods used to generate the prediction sets, while the columns indicate the scoring functions used in the ConfTS optimization process. "↓" indicates smaller values are better. "▲" and "▼" indicate the performance is superior/inferior to the baseline. **Bold** numbers are superior results. Results show that using the non-randomized APS score achieves the overall best performance.

| Model | Score | Base | | APS_no_random | | RAPS_no_random | | APS_random | | RAPS_random | |
|---|---|---|---|---|---|---|---|---|---|---|---|
| | | Avg.size ↓ | Cov. | Avg.size ↓ | Cov. | Avg.size ↓ | Cov. | Avg.size ↓ | Cov. | Avg.size ↓ | Cov. |
| ResNet18 | APS | 14.09 | 0.900 | **7.531 ▲** | 0.900 | 7.752 ▲ | 0.900 | 13.67 ▲ | 0.900 | 13.97 ▲ | 0.900 |
| | RAPS | 9.605 | 0.900 | **5.003 ▲** | 0.900 | 5.346 ▲ | 0.900 | 11.36 ▼ | 0.900 | 11.58 ▼ | 0.900 |
| ResNet50 | APS | 9.062 | 0.900 | **4.791 ▲** | 0.900 | 5.201 ▲ | 0.900 | 12.92 ▼ | 0.900 | 16.43 ▼ | 0.900 |
| | RAPS | 5.992 | 0.900 | **3.561 ▲** | 0.900 | 3.782 ▲ | 0.900 | 9.838 ▼ | 0.900 | 11.70 ▼ | 0.900 |

examples should obtain smaller sets than hard ones. In this part, we examine the impact of ConfTS on the adaptiveness of prediction sets and measure the instance difficulty by the order of the ground truth $o(y, \pi(x))$. Specifically, we partition the sample by label order: 1, 2-3, 4-6, 7-10, 11-100, 101-1000, following (Angelopoulos et al., 2021b). Figure 3a and Figure 3b present the adaptiveness performance of ConfTS with APS score, using ResNet18 and ResNet50 on the ImageNet at $\alpha = 0.1$. A salient observation is that prediction sets, when applied with ConfTS, satisfy the adaptiveness property. Notably, employing ConfTS can promote smaller prediction sets for all examples ranging from easy to hard. Overall, the results demonstrate that APS with ConfTS succeeds in producing adaptive prediction sets: examples with lower difficulty obtain smaller prediction sets on average.

**Ablation study on the size of validation and calibration set.** In the experiment, ConfTS splits the calibration data into two subsets: validation set for tuning the temperature and conformal set for conformal calibration. In this part, we analyze the impact of this split on the performance of ConfTS by varying the validation and conformal dataset sizes from 3,000 to 8,000 samples while maintaining the other part at 5,000 samples. We use ResNet18 and ResNet50 on ImageNet, with APS at $\alpha = 0.1$. Figure 3c and 3d show that the performance of ConfTS remains consistent across different conformal dataset sizes and validation dataset sizes. Based on these results, we choose a calibration set including 10000 samples and split it into two equal subsets for the validation and conformal set. In summary, the performance of ConfTS is robust to variations in the validation dataset and conformal dataset size.

**Ablation study on the non-conformity score in ConfTS.** In this ablation, we compare the performance of ConfTS trained with various non-conformity scores in Eq. (8), including standard APS and RAPS, as well as their non-randomized variants. Table 4 presents the performance of prediction sets generated by standard APS and RAPS ($\lambda = 0.001$) methods with different variants of ConfTS, employing ResNet18 and ResNet50 on ImageNet. The results show that ConfTS with randomized scores fails to produce efficient prediction sets, while non-randomized scores result in small prediction sets. This is because the inclusion of the random variable $u$ leads to the wrong

estimation of the efficiency gap, thereby posing challenges to the optimization process in ConfTS. Moreover, randomized APS consistently performs better than randomized RAPS, even in the case of using the standard RAPS to generate prediction sets. Overall, our findings show that ConfTS with the non-randomized APS outperforms the other scores in enhancing the efficiency of prediction sets.

## 6 RELATED WORK

**Conformal prediction.** Conformal prediction (Papadopoulos et al., 2002; Vovk et al., 2005) is a statistical framework for uncertainty qualification. Previous works have applied conformal prediction across various domains, including regression (Lei & Wasserman, 2014; Romano et al., 2019), image classification (Sadinle et al., 2019; Angelopoulos et al., 2021b; Huang et al., 2024), hyperspectral image classification (Liu et al., 2024), object detection (Angelopoulos et al., 2021a; Teng et al.), and large language models (Kumar et al., 2023).

Some methods leverage post-hoc techniques to enhance the performance of prediction sets (Romano et al., 2020; Angelopoulos et al., 2021b; Ghosh et al., 2023; Huang et al., 2024). For example, Adaptive Prediction Sets (APS) (Romano et al., 2020) calculates the score by accumulating the sorted softmax values in descending order. However, the softmax probabilities typically exhibit a long-tailed distribution, and thus those tail classes are often included in the prediction sets. To alleviate this issue, Regularized Adaptive Prediction Sets (RAPS) (Angelopoulos et al., 2021b) exclude tail classes by appending a penalty to these classes, resulting in efficient prediction sets. Moreover, these post-hoc methods often employ temperature scaling for better calibration performance (Angelopoulos et al., 2021b; Lu et al., 2022; Gibbs et al., 2023; Lu et al., 2023). In our work, we show that the bulk of confidence calibration methods increase the average size of the prediction set, which motivates us to design a variant of temperature scaling, i.e., ConfTS, to enhance the efficiency of prediction sets.

Some works propose training time regularizations to improve the efficiency of conformal prediction (Colombo & Vovk, 2020; Stutz et al., 2022; Einbinder et al., 2022; Bai et al.; Correia et al., 2024). For example, uncertainty-aware conformal loss function (Einbinder et al., 2022) optimizes the efficiency of prediction sets by encouraging the non-conformity scores to follow a uniform distribution. Moreover, conformal training (Stutz et al., 2022) constructs efficient prediction sets by prompting the threshold to be less than the non-conformity scores. In addition, information-based conformal training (Correia et al., 2024) incorporates side information into the construction of prediction sets. In this work, we mainly focus on enhancing adaptive conformal prediction in a post-hoc manner, which is parameter-efficient and requires low computational resources. Notably, our findings show that ConfTS can effectively improve the performance of both post-hoc and training-time conformal prediction methods.

**Confidence calibration.** Confidence calibration has been studied in various contexts in recent years. Some works address the miscalibration problem by post-hoc methods, including histogram binning (Zadrozny & Elkan, 2001) and Platt scaling (Platt et al., 1999). Besides, regularization methods like entropy regularization (Pereyra et al., 2017) and focal loss (Mukhoti et al., 2020) are also proposed to improve the calibration performance of deep neural networks. The most related work explored the influence of confidence calibration on confidence intervals in binary classification settings (Gupta et al., 2020). Our work aligns with and extends these findings by focusing on multi-class classification scenarios, which are more prevalent in practical applications. Moreover, we provide a detailed examination of how the temperature value affects the performance of adaptive conformal prediction.

## 7 CONCLUSION

In this paper, we introduce Conformal Temperature Scaling (ConfTS), a modification to Temperature Scaling that enhances the efficiency of adaptive conformal prediction. ConfTS optimizes the temperature value by minimizing the efficiency gap on a held-out validation set. The obtained temperature would encourage the prediction sets to approximate optimal sets of high efficiency while maintaining the marginal coverage. Extensive experiments show that with ConfTS, the prediction set can be efficient and have better conditional coverage performance. By enhancing adaptive conformal prediction in a post-hoc manner, ConfTS can be easily implemented within any deep learning framework without sacrificing predictive performance. We hope that the insights from this study can serve as a guideline for researchers to effectively incorporate temperature scaling into conformal prediction.

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

## A    CONFORMAL PREDICTION METHODS AND METRICS

In practice, we often use *coverage* and *average size* to evaluate prediction sets, defined as:

$$\text{Coverage} = \frac{1}{|\mathcal{D}_{test}|} \sum_{(\boldsymbol{x}_i, y_i) \in \mathcal{D}_{test}} \mathbb{1}\{y_i \in \mathcal{C}(\boldsymbol{x}_i)\}, \tag{10}$$

$$\text{Average size} = \frac{1}{|\mathcal{D}_{test}|} \sum_{(\boldsymbol{x}_i, y_i) \in \mathcal{D}_{test}} |\mathcal{C}(\boldsymbol{x}_i)|, \tag{11}$$

where $\mathbb{1}$ is the indicator function and $\mathcal{D}_{test}$ denotes the test dataset. The coverage rate measures the percentage of samples whose prediction set contains the true label, i.e., an empirical estimation for $\mathbb{P}(Y \in \mathcal{C}(X))$. Average size measures the efficiency of prediction sets. The prediction sets should both provide valid coverage (defined in Eq. (2)) and efficiency (i.e., small prediction sets). Smaller prediction sets are often preferred since they are more informative in practice (Vovk, 2012; Angelopoulos et al., 2021b).

Moreover, we use *size-stratified coverage violation (SSCV)* (Angelopoulos et al., 2021b) to measure the performance of prediction sets on conditional coverage. Specifically, considering a disjoint set-size strata $\{S_i\}_{i=1}^{N_s}$, where $\bigcup_{i=1}^{N_s} S_i = \{1, 2, \cdots, |\mathcal{Y}|\}$. Then, we define the indexes of examples stratified by the prediction set size by $\mathcal{J}_j = \{i : |\mathcal{C}(\boldsymbol{x}_i)| \in S_j\}$. Formally, we can define the *SSCV* as:

$$\text{SSCV} = \sup_j \left| \frac{|\{i \in \mathcal{J}_j : y_i \in \mathcal{C}(\boldsymbol{x}_i)\}|}{|\mathcal{J}_j|} - (1 - \alpha) \right|. \tag{12}$$

This metric measures the maximum deviation from the target coverage rate $1 - \alpha$ across all strata. A lower SSCV value generally indicates better conditional coverage performance of the prediction sets.

## B    CONFIDENCE CALIBRATION METHODS

Here, we briefly review three post-hoc calibration methods, whose parameters are optimized with respect to negative log-likelihood (NLL) on the calibration set, and three training calibration methods. Let $\sigma$ be the softmax function and $\boldsymbol{f} \in \mathbb{R}^K$ be an arbitrary logits vector.

**Platt Scaling (Platt et al., 1999)**    is a parametric approach for calibration. Platt Scaling learns two scalar parameters $a, b \in \mathbb{R}$ and outputs

$$\pi = \sigma(a\boldsymbol{f} + b). \tag{13}$$

**Temperature Scaling (Guo et al., 2017)**    is inspired by Platt scaling (Platt et al., 1999), using a scalar parameter $t$ for all logits vectors. Formally, for any given logits vector $\boldsymbol{f}$, the new prediction is defined by

$$\pi = \sigma(\boldsymbol{f}/t).$$

Intuitively, $t$ softens the softmax probabilities when $t > 1$ so that it alleviates over-confidence.

**Vector Scaling (Guo et al., 2017)**    is a simple extension of Platt scaling. Let $\boldsymbol{f}$ be an arbitrary logit vector, which is produced before the softmax layer. Vector scaling applies a linear transformation:

$$\pi = \sigma(M\boldsymbol{f} + b),$$

where $M \in \mathbb{R}^{K \times K}$ and $b \in \mathbb{R}^K$.

**Label Smoothing (Szegedy et al., 2016)**    softens hard labels with an introduced smoothing parameter $\alpha$ in the standard loss function (e.g., cross-entropy):

$$\mathcal{L} = -\sum_{k=1}^{K} y_i^* \log p_i, \quad y_k^* = y_k(1 - \alpha) + \alpha/K,$$

where $y_k$ is the soft label for $k$-th class. It is shown that label smoothing encourages the differences between the logits of the correct class and the logits of the incorrect class to be a constant depending on $\alpha$.

**Mixup (Zhang et al., 2018)** is another classical work in the line of training calibration. Mixup generates synthetic samples during training by convexly combining random pairs of inputs and labels as well. To mix up two random samples $(x_i, y_i)$ and $(x_j, y_j)$, the following rules are used:

$$\bar{x} = \alpha x_i + (1 - \alpha) x_j, \quad \bar{y} = \alpha y_i + (1 - \alpha) y_j,$$

where $(\bar{x}_i, \bar{y}_i)$ is the virtual feature-target of original pairs. Previous work (Thulasidasan et al., 2019) observed that compared to the standard models, mixup-trained models are better calibrated and less prone to overconfidence in prediction on out-of-distribution and noise data.

**Bayesian Method (Daxberger et al., 2021).** Bayesian modeling provides a principled and unified approach to mitigate poor calibration and overconfidence by equipping models with robust uncertainty estimates. Specifically, Bayesian modeling handles uncertainty in neural networks by modeling the distribution over the weights. In this approach, given observed data $\mathcal{D} = \{X, y\}$, we aim to infer a posterior distribution over the model parameters $\theta$ using Bayes' theorem:

$$p(\theta|\mathcal{D}) = \frac{p(\mathcal{D}|\theta)p(\theta)}{p(\mathcal{D})}. \tag{14}$$

Here, $p(\mathcal{D}|\theta)$ represents the likelihood, $p(\theta)$ is the prior over the model parameters, and $p(\mathcal{D})$ is the evidence (marginal likelihood). However, the exact posterior $p(\theta|\mathcal{D})$ is often intractable for deep neural networks due to the high-dimensional parameter space, which makes approximate inference techniques necessary.

One common method for approximating the posterior is *Laplace approximation* (LA). The Laplace approximation assumes that the posterior is approximately Gaussian in the vicinity of the optimal parameters $\theta_{\text{MAP}}$, which simplifies inference. Mathematically, LA begins by finding the MAP estimate:

$$\theta_{\text{MAP}} = \arg\max_{\theta} \log p(\mathcal{D}|\theta) + \log p(\theta). \tag{15}$$

Then, the posterior is approximated by a Gaussian distribution:

$$p(\theta|\mathcal{D}) \approx \mathcal{N}(\theta_{\text{MAP}}, H^{-1}), \quad H = -\nabla_{\theta}^2 \log p(\theta|\mathcal{D})\Big|_{\theta=\theta_{\text{MAP}}}. \tag{16}$$

The LA provides an efficient and scalable method to capture uncertainty around the MAP estimate, making it a widely used baseline in Bayesian deep learning models.

## C EXPERIMENTAL SETUP OF SECTION 3.1

**Datasets.** We use two datasets in our study: ImageNet (Deng et al., 2009) and CIFAR-100 (Krizhevsky et al., 2009). On ImageNet, we split the test dataset including 50,000 images into 10,000 images for the calibration set and 40,000 images for the test set; on CIFAR-100, we split the test dataset including 10,000 images into 4,000 images for the calibration set and 6,000 for the test set. Then, we split the calibration set into two subsets of equal size: one is the validation set used for confidence calibration, while the other half is the conformal set used for conformal calibration.

**Models.** We employ three pre-trained classifiers: ResNet18, ResNet50, ResNet101 (He et al., 2016) from TorchVision (Paszke et al., 2019); on CIFAR-100, we train models from scratch. The models are trained for 100 epochs using SGD with a momentum of 0.9, a weight decay of 0.0005, and a batch size of 258. We set the initial learning rate as 0.1 and reduce it by a factor of 5 at 60 epochs. For ConfTr, we set the smoothing parameter $T = 0.1$, penalty term $\kappa = 1$, and hyperparameter $\lambda = 1$, using the same setups. We conduct all the experiments on NVIDIA GeForce RTX 4090.

**Conformal prediction algorithms.** We leverage APS and RAPS to generate prediction sets at an error rate $\alpha = 0.1$, and the hyperparameters are set to be $k_{reg} = 1$ and $\lambda = 0.001$.

**Evaluation.** For the evaluation metrics, we employ *coverage* and *average size* (see Appendix A) to evaluate the performance of prediction sets and utilize ECE to measure the miscalibration (see Section 3.1). All experiments are repeated 20 times with different seeds, and we report the average performance.

# D    RESULTS OF TRAINING-TIME CALIBRATION METHODS

In this section, we report the results of how training-time calibration methods and Bayesian deep learning with Laplace approximation affect the conformal prediction methods. Specifically, we employ label smoothing and mixup to train a ResNet50 model on the CIFAR100 dataset from scratch and utilize Laplace approximation in a post-hoc manner. For label smoothing, we set the hyperparameter $\alpha = 0.05$, and for mixup, the hyperparameter is set to be $\alpha = 0.1$. The training details are available in Appendix C. In Table 5, we show that employing these calibration methods enlarges the prediction sets of APS and RAPS, which is consistent with our results in the main paragraph. For example, with label smoothing, the average size of APS increases from 4.91 to 11.9.

Table 5: Results of different calibration methods using ResNet50 on CIFAR-100. "↓" indicates smaller values are better. "▲" and "▼" indicate whether the performance is superior/inferior to the baseline. **Bold** numbers are superior results. Results show that existing training-time calibration methods and Bayesian deep learning often hurt the efficiency of adaptive conformal prediction.

| Method | $\alpha = 0.1$ | | | | $\alpha = 0.05$ | | | |
|---|---|---|---|---|---|---|---|---|
| | Baseline | LabelSmoothing | Mixup | Bayesian | Baseline | LabelSmoothing | Mixup | Bayesian |
| Accuracy | 0.77 | 0.78 | 0.78 | 0.77 | 0.77 | 0.78 | 0.78 | 0.77 |
| ECE(%) ↓ | 8.79 | 4.39 ▲ | 2.96 ▲ | 4.3 ▲ | 8.79 | 4.39 ▲ | 2.96 ▲ | 4.3 ▲ |
| Avg.size(APS) ↓ | **4.91** | 11.9 ▼ | 12.5 ▼ | 7.55 ▼ | **11.1** | 19.8 ▼ | 20.1 ▼ | 15.6 ▼ |
| Coverage(APS) | 0.90 | 0.90 | 0.90 | 0.90 | 0.95 | 0.95 | 0.95 | 0.95 |
| Avg.size(RAPS) ↓ | **2.56** | 9.5 ▼ | 10.2 ▼ | 6.46 ▼ | **6.95** | 14.5 ▼ | 15.5 ▼ | 9.34 ▼ |
| Coverage(RAPS) | 0.90 | 0.90 | 0.90 | 0.90 | 0.95 | 0.95 | 0.95 | 0.95 |

# E    EXPERIMENTAL SETUP OF SECTION 3.2

In the previous section, we empirically show that current confidence calibration methods negatively impact the efficiency of prediction sets. This motivates our investigation of the impact of high-confidence prediction on prediction set efficiency. The analysis is conducted on the ImageNet dataset with various model architectures, using APS and RAPS at $\alpha = 0.1$. We employ temperature scaling (Guo et al., 2017) in the experiment as it is the simplest method to adjust the confidence level with only a temperature parameter $T$. Specifically, as proven in Lemma G.1(2), lower temperature values consistently encourage higher confidence predictions. This enables us to provide a thorough analysis with theoretical and empirical results, revealing the relationship between confidence calibration and conformal prediction.

# F    WHY NUMERICAL ERROR OCCURS UNDER AN EXCEEDINGLY SMALL TEMPERATURE?

In Section 3.3, we show that an exceedingly low temperature could pose challenges for prediction sets. This problem can be attributed to numerical errors. Specifically, in Proposition 3.1, we show that the softmax probability tends to concentrate in top classes with a small temperature, resulting in a long-tail distribution. Thus, the tail probabilities of some samples could be small and truncated, eventually becoming zero. For example, in Figure 4, the softmax probability is given by $\boldsymbol{\pi}(\boldsymbol{x}) = [0.999997, 2 \times 10^{-5}, 1 \times 10^{-6}, \cdots]$, and the prediction set size should be 4, following Eq. (4). However, due to numerical error, the tail probabilities, i.e., $\pi_5, \pi_6$ are truncated to be zero. This numerical error causes the conformal threshold to exceed the non-conformity scores for all classes, leading to a trivial set. Furthermore, as the temperature decreases, numerical errors occur in more data samples, resulting in increased trivial sets and consequently raising the average set size.

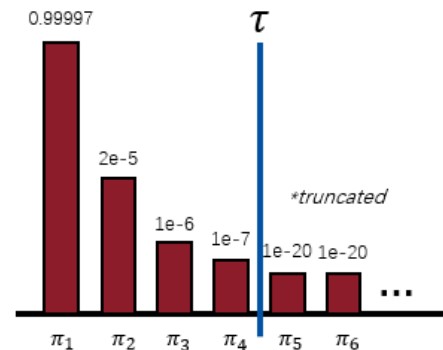

Figure 4: An example of softmax probabilities produced by a small temperature.

# G PROOFS

## G.1 PROOF FOR PROPOSITION 3.1

We start by showing several lemmas: the Lemma G.1, Lemma G.2 and Lemma G.3.

**Lemma G.1.** *For any given logits* $(f_1, \cdots, f_K)$ *with* $f_1 > f_2 > \cdots > f_K$, *and a constant* $0 < t < 1$, *we have:*

$$(a) \quad \frac{e^{f_1/t}}{\sum_{i=1}^{K} e^{f_i/t}} > \frac{e^{f_1}}{\sum_{i=1}^{K} e^{f_i}},$$

$$(b) \quad \frac{e^{f_K/t}}{\sum_{i=1}^{K} e^{f_i/t}} < \frac{e^{f_K}}{\sum_{i=1}^{K} e^{f_i}}.$$

*Proof.* Let $s = \frac{1}{t} - 1$. Then, we have

$$\frac{e^{f_1/t}}{\sum_{i=1}^{K} e^{f_i/t}} = \frac{e^{(1+s)f_1}}{\sum_{i=1}^{K} e^{(1+s)f_i}} = \frac{e^{f_1}}{\sum_{i=1}^{K} e^{f_i} e^{s(f_i-f_1)}} > \frac{e^{f_1}}{\sum_{i=1}^{K} e^{f_i}}.$$

$$\frac{e^{f_K/t}}{\sum_{i=1}^{K} e^{f_i/t}} = \frac{e^{(1+s)f_K}}{\sum_{i=1}^{K} e^{(1+s)f_i}} = \frac{e^{f_K}}{\sum_{i=1}^{K} e^{f_i} e^{s(f_i-f_K)}} < \frac{e^{f_1}}{\sum_{i=1}^{K} e^{f_i}}.$$

$\square$

**Lemma G.2.** *For any given logits* $(f_1, \cdots, f_K)$ *with* $f_1 > f_2 > \cdots > f_K$, *and a constant* $0 < t < 1$, *if there exists* $j > 1$ *such that*

$$\frac{e^{f_j/t}}{\sum_{i=1}^{K} e^{f_i/t}} > \frac{e^{f_j}}{\sum_{i=1}^{K} e^{f_i}},$$

*then, for all* $k = 1, 2, \cdots, j$, *we have*

$$\frac{e^{f_k/t}}{\sum_{i=1}^{K} e^{f_i/t}} > \frac{e^{f_k}}{\sum_{i=1}^{K} e^{f_i}}. \tag{17}$$

*Proof.* It suffices to show that

$$\frac{e^{f_{j-1}/t}}{\sum_{i=1}^{K} e^{f_i/t}} > \frac{e^{f_{j-1}}}{\sum_{i=1}^{K} e^{f_i}}, \tag{18}$$

since the rest cases where $k = 1, 2, \cdots, j-1$ would hold by induction. The assumption gives us

$$\frac{e^{f_j/t}}{\sum_{i=1}^{K} e^{f_i/t}} > \frac{e^{f_j}}{\sum_{i=1}^{K} e^{f_i}}.$$

Let $s = \frac{1}{t} - 1$, which follows that

$$\frac{e^{f_j/t}}{\sum_{i=1}^{K} e^{f_i/t}} = \frac{e^{(1+s)f_j}}{\sum_{i=1}^{K} e^{(1+s)f_i}} = \frac{e^{f_j}}{\sum_{i=1}^{K} e^{f_i} e^{s(f_i - f_j)}} \overset{(a)}{>} \frac{e^{f_j}}{\sum_{i=1}^{K} e^{f_i}}.$$

The inequality (a) indicates that

$$\sum_{i=1}^{K} e^{f_i} e^{s(f_i - f_j)} < \sum_{i=1}^{K} e^{f_i}.$$

Therefore, we can have

$$\frac{e^{f_{j-1}/t}}{\sum_{i=1}^{K} e^{f_i/t}} = \frac{e^{(1+s)f_{j-1}}}{\sum_{i=1}^{K} e^{(1+s)f_i}} = \frac{e^{f_{j-1}}}{\sum_{i=1}^{K} e^{f_i} e^{s(f_i - f_{j-1})}} > \frac{e^{f_{j-1}}}{\sum_{i=1}^{K} e^{f_i} e^{s(f_i - f_j)}} > \frac{e^{f_{j-1}}}{\sum_{i=1}^{K} e^{f_i}},$$

which proves the Eq. (18). Then, by induction, the Eq. (17) holds for all $1 \leq k < j$. $\qquad\square$

**Lemma G.3.** *For any given logits $(f_1, \cdots, f_K)$, where $f_1 > f_2 > \cdots > f_K$, a constant $0 < t < 1$, and for all $k = 1, 2, \cdots, K$, we have*

$$\sum_{i=1}^{k} \frac{e^{f_i/t}}{\sum_{j=1}^{K} e^{f_j/t}} \geq \sum_{i=1}^{k} \frac{e^{f_i}}{\sum_{j=1}^{K} e^{f_j}} \tag{19}$$

*The equation holds if and only if $k = K$.*

*Proof.* The Eq. (19) holds trivially at $k = K$, since both sides are equal to 1:

$$\sum_{i=1}^{K} \frac{e^{f_i/t}}{\sum_{j=1}^{K} e^{f_j/t}} = \sum_{i=1}^{K} \frac{e^{f_i}}{\sum_{j=1}^{K} e^{f_j}} = 1, \tag{20}$$

We continue by showing the Eq. (19) at $k = K - 1$. The Lemma G.1 gives us that

$$\frac{e^{f_K/t}}{\sum_{i=1}^{K} e^{f_i/t}} < \frac{e^{f_K}}{\sum_{i=1}^{K} e^{f_i}}, \tag{21}$$

Subtracting the Eq. (21) by the Eq. (21) directly follows that

$$\sum_{i=1}^{K-1} \frac{e^{f_i/t}}{\sum_{j=1}^{K} e^{f_j/t}} > \sum_{i=1}^{K-1} \frac{e^{f_i}}{\sum_{j=1}^{K} e^{f_j}}, \tag{22}$$

which prove the Eq. (19) at $k = K - 1$. We then show that the Eq. (19) holds at $k = K - 2$, which follows that the Eq. (19) remains true for all $k = 1, 2, \cdots K - 1$ by induction. Here, we assume that

$$\sum_{i=1}^{K-2} \frac{e^{f_i/t}}{\sum_{j=1}^{K} e^{f_j/t}} < \sum_{i=1}^{K-2} \frac{e^{f_i}}{\sum_{j=1}^{K} e^{f_j}}, \tag{23}$$

and we will show that the Eq. (23) leads to a contradiction. Subtracting Eq. (23) by the Eq. (22) gives us that

$$\frac{e^{f_{K-1}/t}}{\sum_{i=1}^{K} e^{f_i/t}} > \frac{e^{f_{K-1}}}{\sum_{i=1}^{K} e^{f_i}}. \tag{24}$$

Considering the Lemma G.2, the Eq. (24) implies that

$$\frac{e^{f_k/t}}{\sum_{i=1}^{K} e^{f_i/t}} > \frac{e^{f_k}}{\sum_{i=1}^{K} e^{f_i}} \tag{25}$$

holds for all $k = 1, 2, \cdots, K - 2$. Accumulating the Eq. (25) from $k = 1$ to $K - 2$ gives us that

$$\sum_{i=1}^{K-2} \frac{e^{f_i/t}}{\sum_{j=1}^{K} e^{f_j/t}} > \sum_{i=1}^{K-2} \frac{e^{f_i}}{\sum_{j=1}^{K} e^{f_j}}. $$

This contradicts our assumption (Eq. (23)). It follows that Eq. (19) holds at $k = K - 2$. Then, by induction, the Eq. (19) remains true for all $k = 1, 2, \cdots K - 1$. Combining with the Eq. (20), we can complete our proof. $\qquad\square$

**Proposition G.4** (Restatement of Proposition 3.1). *For any sample $\boldsymbol{x} \in \mathcal{X}$, let $\mathcal{S}(\boldsymbol{x}, k, t)$ be the non-conformity score function with respect to an arbitrary class $k \in \mathcal{Y}$, defined as in Eq. 7. Then, for a fixed temperature $t_0$ and $\forall t \in (0, t_0)$, we have*

$$\mathcal{S}(\boldsymbol{x}, k, t_0) \leq \mathcal{S}(\boldsymbol{x}, k, t).$$

*Proof.* We restate the definition of non-randomized APS score in Eq. 7:

$$\mathcal{S}(\boldsymbol{x}, y, t) = \sum_{i=1}^{k} \frac{e^{f_i}}{\sum_{j=1}^{K} e^{f_j}}.$$

Let $\alpha = t/t_0 \in (0, 1)$ and $\tilde{f}_i = f_i/t_0$. We rewrite the formulation of $\mathcal{S}(\boldsymbol{x}, k, t_0)$ and $\mathcal{S}(\boldsymbol{x}, k, t)$ by

$$\mathcal{S}(\boldsymbol{x}, y, t_0) = \sum_{i=1}^{k} \frac{e^{\tilde{f}_i}}{\sum_{j=1}^{K} e^{\tilde{f}_j}},$$

$$\mathcal{S}(\boldsymbol{x}, y, t) = \sum_{i=1}^{k} \frac{e^{\tilde{f}_i/\alpha}}{\sum_{j=1}^{K} e^{\tilde{f}_j/\alpha}}.$$

Since the scaling parameter $t_0$ does not change the order of $(\tilde{f}_1, \tilde{f}_2, \cdots, \tilde{f}_K)$, i.e. $\tilde{f}_1 > \tilde{f}_2 > \cdots > \tilde{f}_K$ and $\alpha \in (0, 1)$, then by the Lemma G.3, we have $\mathcal{S}(\boldsymbol{x}, y, t_0) < \mathcal{S}(\boldsymbol{x}, y, t)$. $\square$

## G.2 PROOF FOR COROLLARY 3.2

**Corollary G.5** (Restatement of Corollary 3.2). *For any sample $\boldsymbol{x} \in \mathcal{X}$ and a fixed temperature $t_0$, the difference $\epsilon(k, t)$ is a decreasing function with respect to $t \in (0, t_0)$.*

*Proof.* For all $t_1, t_2$ satisfying $0 < t_1 < t_2 < t_0$, we will show that $\epsilon(k, t_1) > \epsilon(k, t_2)$. Continuing from Proposition 3.1, we have $\mathcal{S}(\boldsymbol{x}, y, t_2) < \mathcal{S}(\boldsymbol{x}, y, t_1)$. It follows that

$$\begin{aligned}
\epsilon(k, t_1) &= \mathcal{S}(\boldsymbol{x}, k, t_1) - \mathcal{S}(\boldsymbol{x}, k, t_0) \\
&> \mathcal{S}(\boldsymbol{x}, k, t_2) - \mathcal{S}(\boldsymbol{x}, k, t_0) \\
&= \epsilon(k, t_2).
\end{aligned}$$

$\square$

## G.3 PROOF FOR THEOREM 3.3

In the theorem, we make two continuity assumptions on the CDF of the non-conformity score following (Lei, 2014; Sadinle et al., 2019). We define $G_k^t(\cdot)$ as the CDF of $\mathcal{S}(\boldsymbol{x}, k, t)$, assuming that

$$\begin{aligned}
&(1) \exists \gamma, c_1, c_2 \in (0, 1] \; s.t. \; \forall k \in \mathcal{Y}, \; c_1 |\varepsilon|^{\gamma} \leq |G_k^t(s + \varepsilon) - G_k^t(s)| \leq c_2 |\varepsilon|^{\gamma}, \\
&(2) \exists \rho > 0 \; s.t. \; \inf_{k, s} |G_k^{t_0}(s) - G_k^t(s)| \geq \rho.
\end{aligned} \tag{26}$$

To prove Theorem 3.3, we start with a lemma:

**Lemma G.6.** *Give a pre-trained model, data sample $\boldsymbol{x}$, and a temperature satisfying $t^* < t_0$. Then, under assumtion (26), we have*

$$\mathbb{P}\{k \in \mathcal{C}(\boldsymbol{x}, t_0), k \notin \mathcal{C}(\boldsymbol{x}, t^*)\} \geq c_1 (2\epsilon(k, t^*))^{\gamma}.$$

*Proof.* Let $\mathbb{P}^t(\cdot)$ be the probability measure corresponding to $G_y^t(\cdot)$, and $C_y^t(s) = \{x : S(\boldsymbol{x}, y, t) < s\}$. Then, we have

$$\begin{aligned}
\mathbb{P}^{t_0}(C_y^{t_0}(\tau(t^*))) &= \mathbb{P}^{t_0}(C_y^{t^*}(\tau(t^*) + \epsilon(k, t^*))) \\
&= G_y^{t_0}(\tau(t^*) + \epsilon(k, t^*)) \\
&\overset{(a)}{\geq} G_y^{t^*}(\tau(t^*) + \epsilon(k, t^*)) + \rho.
\end{aligned} \tag{27}$$

where (a) comes from the assumption (2). Let $\tau^* = \tau(t^*) - \epsilon(k, t^*) - [c_2^{-1}\rho]^{1/\gamma}$. Then, replacing the $\tau(t^*)$ in Eq. (27) with $\tau^*$, we have

$$
\begin{aligned}
\mathbb{P}^{t_0}(C_y^{t_0}(\tau^*)) &\geq G_y^{t^*}(\tau(t^*) - [c_2^{-1}\rho]^{1/\gamma}) + \rho \\
&\stackrel{(a)}{\geq} G_y^{t^*}(\tau(t^*)) \\
&\stackrel{(b)}{=} \alpha \\
&\stackrel{(c)}{=} \mathbb{P}^{t_0}(C_y^{t_0}(\tau(t_0))).
\end{aligned}
\tag{28}
$$

where (a) is due to the assumption (1):

$$
G_y^{t^*}(\tau(t^*)) - G_y^{t^*}(\tau(t^*) - [c_2^{-1}\rho]^{1/\gamma}) \leq c_2|[c_2^{-1}\rho]^{1/\gamma}|^\gamma = \rho.
$$

(b) and (c) is because of the definition of threshold $\tau$: $C_y^{t^*}(\tau(t^*)) = C_y^{t_0}(\tau(t_0)) = \alpha$. The Eq. (29) follows that

$$
\tau(t_0) \leq \tau^* = \tau(t^*) - \epsilon(k, t^*) - [c_2^{-1}\rho]^{1/\gamma}.
\tag{29}
$$

Continuing from Eq. (29), it holds for all $y \in \mathcal{Y}$ that

$$
\begin{aligned}
\mathbb{P}\{k \in \mathcal{C}(\boldsymbol{x}, t_0), k \notin \mathcal{C}(\boldsymbol{x}, t^*)\} &\stackrel{(a)}{=} P\{\mathcal{S}(\boldsymbol{x}, y, t^*) < \tau(t^*), \mathcal{S}(\boldsymbol{x}, y, t_0) \geq \tau(t_0)\} \\
&\stackrel{(b)}{=} P\{\tau(t^*) > S(\boldsymbol{x}, y, t^*) \geq \tau(t_0) - \epsilon(k, t^*)\} \\
&\geq P\{\tau(t^*) > S(\boldsymbol{x}, y, t^*) \geq \tau(t^*) - 2\epsilon(k, t^*) - [c_2^{-1}\rho]^{1/\gamma}\} \\
&\stackrel{(c)}{=} G_y^{t^*}(\tau(t^*)) - G_y^{t^*}(\tau(t^*) - 2\epsilon(k, t^*) - [c_2^{-1}\rho]^{1/\gamma}) \\
&\stackrel{(d)}{\geq} c_1(2\epsilon(k, t^*) + [c_2^{-1}\rho]^{1/\gamma})^\gamma \\
&\geq c_1(2\epsilon(k, t^*))^\gamma.
\end{aligned}
$$

where (a) comes from the construction of prediction set: $y \in \mathcal{C}(\boldsymbol{x})$ if and only if $\mathcal{S}(\boldsymbol{x}, y) \leq \tau$. (b) is because of the definition of $\epsilon$. (c) and (d) is due to the definition of $G_y^t(\cdot)$ and assumption (1). $\qquad\square$

**Theorem G.7.** *Under the assumption* (26)*, there exists constants* $c_1, \gamma \in (0, 1]$ *such that*

$$
\mathbb{E}_{\boldsymbol{x} \in \mathcal{X}}[|\mathcal{C}(\boldsymbol{x}, t)|] \leq K - \sum_{k \in \mathcal{Y}} c_1[2\epsilon(k, t)]^\gamma, \quad \forall t \in (0, t_0).
$$

*Proof.* For all $t < t_0$, we consider the expectation size of $\mathcal{C}(\boldsymbol{x}, t)$:

$$
\begin{aligned}
\mathbb{E}_{x \in \mathcal{X}}[|\mathcal{C}(\boldsymbol{x}, t)|] &= \mathbb{E}_{x \in \mathcal{X}}\left[\sum_{k \in \mathcal{Y}} \mathbb{1}\{k \in \mathcal{C}(\boldsymbol{x}, t)\}\right] \\
&= \sum_{k \in \mathcal{Y}} \mathbb{E}_{x \in \mathcal{X}}[\mathbb{1}\{k \in \mathcal{C}(\boldsymbol{x}, t)\}] \\
&= \sum_{k \in \mathcal{Y}} \mathbb{P}\{k \in \mathcal{C}(\boldsymbol{x}, t)\} \\
&= \sum_{k \in \mathcal{Y}} [1 - \mathbb{P}\{k \notin \mathcal{C}(\boldsymbol{x}, t)\}].
\end{aligned}
$$

Due to the fact that

$$
\mathbb{P}\{k \in \mathcal{C}(\boldsymbol{x}, t_0), k \notin \mathcal{C}(\boldsymbol{x}, t)\} \leq \mathbb{P}\{k \notin \mathcal{C}(\boldsymbol{x}, t)\},
$$

we have

$$
\mathbb{E}_{x \in \mathcal{X}}[|\mathcal{C}(\boldsymbol{x}, t)|] \leq \sum_{k \in \mathcal{Y}} [1 - \mathbb{P}\{k \in \mathcal{C}(\boldsymbol{x}, t_0), k \notin \mathcal{C}(\boldsymbol{x}, t)\}].
$$

Continuing from Lemma G.6, we can get

$$
\mathbb{E}_{x \in \mathcal{X}}[|\mathcal{C}(\boldsymbol{x}, t)|] \leq K(1 - c_1(2\epsilon(k, t))^\gamma) = K - \sum_{k \in \mathcal{Y}} c_1(2\epsilon(k, t))^\gamma.
$$

$\qquad\square$

### G.4 PROOF FOR PROPOSITION L.3

**Proposition G.8** (Restatement of Proposition L.3). *Define $\tau$ as the $1 - \alpha$ conformal threshold (see Eq. (3) in the main paragraph). Then, we have*

$$\text{MSCE} \leq 2L\gamma \cdot \mathbb{E}_{(x,y)\sim\mathcal{X}\times\mathcal{Y}} \left[ \sqrt{\mathcal{L}_{\text{ConfTS}}(x,y;t)} \right]$$

*where $\gamma = \max(\alpha, 1 - \alpha)$.*

*Proof.* Continuing from the Proposition 3.5 in (Kiyani et al., 2024b), we have

$$\begin{aligned}
\text{MSCE} &\leq 2L \cdot \mathbb{E}_S \left[ l_{1-\alpha}(\tau(t), S) - l_{1-\alpha}(q_{1-\alpha}(X), S) \right] \\
&\leq 2L \cdot \mathbb{E}_S \left[ l_{1-\alpha}(\tau, S) \right] \\
&\leq 2L \cdot \mathbb{E}_S[\max(\alpha, 1 - \alpha)|\tau - S|] \\
&= 2L\gamma \cdot \mathbb{E}_{(x,y)\sim\mathcal{X}\times\mathcal{Y}} \left[ \sqrt{\mathcal{L}_{\text{ConfTS}}(x,y;t)} \right]
\end{aligned}$$

$\square$

## H RESULTS OF CONFTS ON CIFAR-100

In this section, we show that ConfTS can effectively improve the efficiency of adaptive conformal prediction on the CIFAR100 dataset. In particular, we train ResNet18, ResNet50, ResNet191, ResNext50, ResNext101, DenseNet121 and VGG16 from scratch on CIFAR-100 datasets. We leverage APS and RAPS to generate prediction sets at error rates $\alpha \in \{0.1, 0.05\}$. The hyper-parameter for RAPS is set to be $k_{reg} = 1$ and $\lambda = 0.001$. In Table 6, results show that after being employed with ConfTS, APS, and RAPS tend to construct smaller prediction sets and maintain the desired coverage.

Table 6: Performance comparison of the baseline and ConfTS on CIFAR-100 dataset. We employ five models trained on CIFAR-100. "*" denotes significant improvement (two-sample t-test at a 0.1 confidence level). "↓" indicates smaller values are better. **Bold** numbers are superior results. Results show that our ConfTS can improve the performance of APS and RAPS, maintaining the desired coverage rate.

| Model | Score | $\alpha = 0.1$ | | $\alpha = 0.05$ | |
|---|---|---|---|---|---|
| | | Coverage | Average ↓ size | Coverage | Average size ↓ |
| | | Baseline / ConfTS | | | |
| ResNet18 | APS | 0.902 / 0.901 | 7.049 / **6.547***  | 0.949 / 0.949 | 12.58 / **11.91*** |
| | RAPS | 0.900 / 0.901 | 5.745 / **4.948*** | 0.949 / 0.949 | 8.180 / **7.689*** |
| ResNet50 | APS | 0.901 / 0.900 | 5.614 / **5.322*** | 0.951 / 0.951 | 10.27 / **10.00*** |
| | RAPS | 0.900 / 0.900 | 4.707 / **4.409*** | 0.951 / 0.950 | 7.041 / **6.811*** |
| ResNet101 | APS | 0.900 / 0.900 | 5.049 / **4.917*** | 0.949 / 0.949 | 9.520 / **9.405*** |
| | RAPS | 0.901 / 0.900 | 4.324 / **4.145*** | 0.950 / 0.950 | 6.515 / **6.450*** |
| ResNext50 | APS | 0.900 / 0.900 | 4.668 / **4.436*** | 0.950 / 0.950 | 8.911 / **8.626*** |
| | RAPS | 0.901 / 0.901 | 4.050 / **3.811*** | 0.951 / 0.951 | 6.109 / **5.854*** |
| ResNext101 | APS | 0.900 / 0.900 | 4.125 / **3.988*** | 0.950 / 0.950 | 7.801 / **7.614*** |
| | RAPS | 0.901 / 0.901 | 3.631 / **3.492*** | 0.950 / 0.950 | 5.469 / **5.253*** |
| DenseNet121 | APS | 0.899 / 0.899 | 4.401 / **3.901*** | 0.949 / 0.949 | 8.364 / **7.592*** |
| | RAPS | 0.898 / 0.898 | 3.961 / **3.434*** | 0.950 / 0.949 | 6.336 / **5.222*** |
| VGG16 | APS | 0.900 / 0.900 | 7.681 / **6.658*** | 0.949 / 0.950 | 12.36 / **11.70*** |
| | RAPS | 0.899 / 0.900 | 6.826 / **5.304*** | 0.949 / 0.949 | **10.32*** / 11.70 |

# I  RESULTS OF CONFTS ON IMAGENET-V2

In this section, we show that ConfTS can effectively improve the efficiency of adaptive conformal prediction on the ImageNet-V2 dataset. In particular, we employ pre-trained ResNet50, DenseNet121, VGG16, and ViT-B-16 on ImageNet. We leverage APS and RAPS to construct prediction sets and the hyper-parameters of RAPS are set to be $k_{reg} = 1$ and $\lambda = 0.001$. In Table 7, results show that after being employed with ConfTS, APS, and RAPS tend to construct smaller prediction sets and maintain the desired coverage.

Table 7: Performance comparison of conformal prediction with baseline and ConfTS under distribution shifts. "*" denotes significant improvement (two-sample t-test at a 0.1 confidence level). "↓" indicates smaller values are better. **Bold** numbers are superior results. Results show that ConfTS can improve the efficiency of APS and RAPS on a new distribution.

| Metrics | ResNet50 | | DenseNet121 | | VGG16 | | ViT | |
|---|---|---|---|---|---|---|---|---|
| | Baseline | ConfTS | Baseline | ConfTS | Baseline | ConfTS | Baseline | ConfTS |
| Avg.size(APS) ↓ | 24.6 | **11.9**\* | 50.3 | **13.3**\* | 27.2 | **17.9**\* | 34.2 | **10.1**\* |
| Coverage(APS) | 0.90 | 0.90 | 0.90 | 0.90 | 0.90 | 0.90 | 0.90 | 0.90 |
| Avg.size(RAPS) ↓ | 13.3 | **11.3**\* | 13.7 | **9.67**\* | 16.3 | **13.6**\* | 14.9 | **4.62**\* |
| Coverage(RAPS) | 0.90 | 0.90 | 0.90 | 0.90 | 0.90 | 0.90 | 0.90 | 0.90 |

# J  RESULTS OF CONFTS ON RAPS WITH VARIOUS PENALTY TERMS

Recall that the RAPS method modifies APS by including a penalty term $\lambda$ (see Eq. (6)). In this section, we investigate the performance of ConfTS on RAPS with various penalty terms. In particular, we employ the same model architectures with the main experiment on ImageNet (see Section 5.1) and generate prediction sets with RAPS ($k_{reg} = 1$) at an error rate $\alpha = 0.1$, varying the penalty $\lambda \in \{0.002, 0.004, 0.006, 0.01, 0.015, 0.02\}$ and setting $k_{reg}$ to 1. Table 8 and 9 show that our ConfTS can enhance the efficiency of RAPS across various penalty values.

Table 8: Performance of ConfTS on RAPS with various penalty terms $\lambda \in \{0.002, 0.004, 0.006\}$ at ImageNet. "*" denotes significant improvement (two-sample t-test at a 0.1 confidence level). "↓" indicates smaller values are better. **Bold** numbers are superior results. Results show that our ConfTS can enhance the efficiency of RAPS across various penalty values.

| Model | $\lambda = 0.002$ | | $\lambda = 0.004$ | | $\lambda = 0.006$ | |
|---|---|---|---|---|---|---|
| | Coverage | Average size ↓ | Coverage | Average size ↓ | Coverage | Average size ↓ |
| | | | Baseline / ConfTS | | | |
| ResNet18 | 0.901 / 0.900 | 8.273 / **4.517**\* | 0.901 / 0.901 | 6.861 / **4.319**\* | 0.901 / 0.901 | 6.109 / **4.282**\* |
| ResNet50 | 0.899 / 0.900 | 5.097 / **3.231**\* | 0.899 / 0.900 | 4.272 / **2.892**\* | 0.899 / 0.900 | 3.858 / **2.703**\* |
| ResNet101 | 0.900 / 0.900 | 4.190 / **2.987**\* | 0.901 / 0.899 | 3.599 / **2.686**\* | 0.900 / 0.900 | 3.267 / **2.516**\* |
| DenseNet121 | 0.901 / 0.901 | 5.780 / **3.340**\* | 0.900 / 0.900 | 4.888 / **3.014**\* | 0.900 / 0.900 | 4.408 / **2.836**\* |
| VGG16 | 0.901 / 0.900 | 7.030 / **3.902**\* | 0.901 / 0.900 | 5.864 / **3.514**\* | 0.901 / 0.900 | 5.241 / **3.344**\* |
| ViT-B-16 | 0.901 / 0.900 | 5.308 / **1.731**\* | 0.901 / 0.901 | 4.023 / **1.655**\* | 0.901 / 0.901 | 3.453 / **1.611**\* |

Table 9: Performance of ConfTS on RAPS with various penalty terms $\lambda \in \{0.01, 0.015, 0.02\}$ at ImageNet. "*" denotes significant improvement (two-sample t-test at a 0.1 confidence level). "↓" indicates smaller values are better. **Bold** numbers are superior results. Results show that our ConfTS can enhance the efficiency of RAPS across various penalty values.

| Model | $\lambda = 0.01$ | | $\lambda = 0.015$ | | $\lambda = 0.02$ | |
|---|---|---|---|---|---|---|
| | Coverage | Average size ↓ | Coverage | Average size ↓ | Coverage | Average size ↓ |
| | Baseline / ConfTS | | | | | |
| ResNet18 | 0.901 / 0.901 | 5.281 / **4.449*** | 0.901 / 0.901 | 4.712 / **4.683*** | 0.900 / 0.900 | **4.452*** / 4.917 |
| ResNet50 | 0.899 / 0.900 | 3.380 / **2.505*** | 0.900 / 0.901 | 3.048 / **2.373*** | 0.901 / 0.901 | 2.860 / **2.321*** |
| ResNet101 | 0.900 / 0.900 | 2.902 / **2.317*** | 0.900 / 0.899 | 2.643 / **2.168*** | 0.900 / 0.900 | 2.484 / **2.096*** |
| DenseNet121 | 0.900 / 0.900 | 3.843 / **2.657*** | 0.900 / 0.900 | 3.452 / **2.587*** | 0.901 / 0.899 | 3.213 / **2.750*** |
| VGG16 | 0.900 / 0.900 | 4.537 / **3.371*** | 0.900 / 0.900 | 4.060 / **3.423*** | 0.899 / 0.899 | 3.744 / **3.530*** |
| ViT-B-16 | 0.901 / 0.900 | 2.872 / **1.564*** | 0.901 / 0.900 | 2.508 / **1.543*** | 0.900 / 0.900 | 2.285 / **1.535*** |

## K  RESULTS OF CONFTS ON SAPS

Recall that APS calculates the non-conformity score by accumulating the sorted softmax values in descending order. However, the softmax probabilities typically exhibit a long-tailed distribution, allowing for easy inclusion of those tail classes in the prediction sets. To alleviate this issue, *Sorted Adaptive Prediction Sets (SAPS)* (Huang et al., 2024) discards all the probability values except for the maximum softmax probability when computing the non-conformity score. Formally, the non-conformity score of SAPS for a data pair $(\boldsymbol{x}, y)$ can be calculated as

$$S_{saps}(\boldsymbol{x}, y, u; \hat{\pi}) := \begin{cases} u \cdot \hat{\pi}_{max}(\boldsymbol{x}), & \text{if } o(y, \hat{\pi}(\boldsymbol{x})) = 1, \\ \hat{\pi}_{max}(\boldsymbol{x}) + (o(y, \hat{\pi}(\boldsymbol{x})) - 2 + u) \cdot \lambda, & \text{else,} \end{cases}$$

where $\lambda$ is a hyperparameter representing the weight of ranking information, $\hat{\pi}_{max}(\boldsymbol{x})$ denotes the maximum softmax probability and $u$ is a uniform random variable.

In this section, we investigate the performance of ConfTS on SAPS with various weight terms. In particular, we employ the same model architectures with the main experiment on ImageNet (see Section 5.1) and generate prediction sets with SAPS at an error rate $\alpha = 0.1$, varying the weight $\lambda \in \{0.01, 0.02, 0.03, 0.05, 0.1, 0.12\}$. Table 10 and Table 12 show that our ConfTS can enhance the efficiency of SAPS across various weights.

Table 10: Performance of ConfTS on SAPS with various penalty terms $\lambda \in [0.005, 0.01, 0.02]$. "*" denotes significant improvement (two-sample t-test at a 0.1 confidence level). "↓" indicates smaller values are better. **Bold** numbers are superior results. Results show that our ConfTS can enhance the efficiency of SAPS across various penalty values.

| Model | $\lambda = 0.005$ | | $\lambda = 0.01$ | | $\lambda = 0.02$ | |
|---|---|---|---|---|---|---|
| | Coverage | Average size ↓ | Coverage | Average size ↓ | Coverage | Average size ↓ |
| | Baseline / ConfTS | | | | | |
| ResNet18 | 0.901 / 0.900 | 37.03 / **27.38*** | 0.901 / 0.902 | 19.91 / **14.81*** | 0.900 / 0.901 | 11.21 / **8.469*** |
| ResNet50 | 0.899 / 0.899 | 27.13 / **21.37*** | 0.899 / 0.899 | 14.45 / **11.48*** | 0.899 / 0.899 | 8.016 / **6.510*** |
| ResNet101 | 0.901 / 0.901 | 24.89 / **20.78*** | 0.901 / 0.901 | 13.21 / **11.16*** | 0.901 / 0.901 | 7.350 / **6.287*** |
| DenseNet121 | 0.900 / 0.901 | 30.54 / **22.67*** | 0.900 / 0.901 | 16.28 / **12.30*** | 0.901 / 0.901 | 9.085 / **6.968*** |
| VGG16 | 0.900 / 0.900 | 34.88 / **25.57*** | 0.900 / 0.900 | 18.56 / **13.71*** | 0.901 / 0.900 | 10.34 / **7.788*** |
| ViT-B-16 | 0.901 / 0.900 | 18.90 / **11.51*** | 0.901 / 0.900 | 10.11 / **6.379*** | 0.900 / 0.900 | 5.669 / **3.784*** |
| Average | 0.900 / 0.900 | 28.89 / **21.54*** | 0.900 / 0.900 | 15.42 / **11.63*** | 0.900 / 0.900 | 8.611 / **6.634*** |

Table 11: Performance of ConfTS on SAPS with various penalty terms $\lambda \in \{0.03, 0.05, 0.1\}$. "*" denotes significant improvement (two-sample t-test at a 0.1 confidence level). "↓" indicates smaller values are better. **Bold** numbers are superior results. Results show that our ConfTS can enhance the efficiency of SAPS across various penalty values.

| Model | $\lambda = 0.03$ | | $\lambda = 0.05$ | | $\lambda = 0.1$ | |
|---|---|---|---|---|---|---|
| | Coverage | Average size ↓ | Coverage | Average size ↓ | Coverage | Average size ↓ |
| | | | Baseline / ConfTS | | | |
| ResNet18 | 0.900 / 0.900 | 8.206 / **6.269**\* | 0.900 / 0.900 | 5.747 / **4.716**\* | 0.901 / 0.901 | **4.143**\* / 4.581 |
| ResNet50 | 0.899 / 0.899 | 5.853 / **4.838**\* | 0.899 / 0.900 | 4.122 / **3.464**\* | 0.899 / 0.900 | 2.753 / **2.460**\* |
| ResNet101 | 0.901 / 0.901 | 5.364 / **4.640**\* | 0.901 / 0.901 | 3.756 / **3.293**\* | 0.899 / 0.900 | 2.511 / **2.286**\* |
| DenseNet121 | 0.900 / 0.900 | 6.600 / **5.151**\* | 0.900 / 0.900 | 4.601 / **3.672**\* | 0.900 / 0.900 | 3.063 / **2.811**\* |
| VGG16 | 0.900 / 0.900 | 7.504 / **5.785**\* | 0.900 / 0.900 | 5.225 / **4.173**\* | 0.900 / 0.900 | **3.483**\* / 3.551 |
| ViT-B-16 | 0.900 / 0.900 | 4.197 / **2.905**\* | 0.900 / 0.900 | 2.995 / **2.212**\* | 0.901 / 0.900 | 2.114 / **1.768**\* |
| Average | 0.900 / 0.900 | 6.287 / **4.931**\* | 0.900 / 0.900 | 4.407 / **3.588**\* | 0.900 / 0.900 | 3.011 / **2.909**\* |

## L   THE PERFORMANCE OF CONFTS ON CONDITIONAL COVERAGE

In this section, we formally analyze why ConfTS enhances conditional coverage for APS. Following previous work (Kiyani et al., 2024b), we use Mean Squared Conditional Error (MSCE) as a measure of conditional coverage performance:

$$\text{MSCE} = \mathbb{E}_{x \sim \mathcal{X}}[\{\text{Coverage}(\mathcal{C}(X)|X = x) - (1 - \alpha)\}^2]$$

In particular, it quantifies how prediction sets deviate from the ideal conditional coverage:

$$\mathbb{P}\{Y \in \mathcal{C}(X)|X = x\} = 1 - \alpha$$

As shown in [1], MSCE is a valid measure of conditional coverage performance. Consider the pinball loss [3]:

$$l_{1-\alpha}(\tau, s) = \alpha(\tau - s)\mathbf{1}\{\tau \geq s\} + (1 - \alpha)(s - \tau)\mathbf{1}\{\tau \leq s\}. \tag{30}$$

where $\mathbf{1}\{\cdot\}$ is the indicator function. Let us now state the required technical assumption:

**Definition L.1.** *A distribution $\mathcal{P}$, is called L-lipschitz if we have for every real numbers $q \leq q'$:*

$$\mathbb{P}_{S \sim \mathcal{P}}\{S \leq q'\} - \mathbb{P}_{S \sim \mathcal{P}}\{S \leq q\} \leq L|q' - q|$$

**Assumption L.2.** *The distribution of $S$ conditional on $X = x$ is L-lipschitz.*

Assumption L.2 is often needed for the analysis of conditional coverage in CP literature in both regression (Jung et al., 2023; Kiyani et al., 2024b) and classification (Kiyani et al., 2024a) setting. In the following theorem, we will show that MSCE can be upper bounded by ConfTS loss:

**Proposition L.3.** *Define $\tau$ as the $1 - \alpha$ conformal threshold (see Eq. (3) in the main paragraph). Then, we have*

$$\text{MSCE} \leq 2L\gamma \cdot \mathbb{E}_{(X,Y) \sim \mathcal{X} \times \mathcal{Y}}\left[\sqrt{\mathcal{L}_{\text{ConfTS}}(X, Y; t)}\right]$$

*where $\gamma = \max(\alpha, 1 - \alpha)$.*

The proof can be found in Appendix G.4. Thus, we conclude that by minimizing $|\tau - S|$ and consequently reducing MSCE, ConfTS improves conditional coverage. The rigorous proofs are available in the supplementary material.

**Important note:** Though ConfTS demonstrates enhanced conditional coverage, we emphasize that this is an auxiliary benefit rather than its core design purpose and we acknowledge that this improvement does not extend to RAPS in terms of SSCV and CSCV. This is because \*\*temperature tuning alone provides limited capacity for minimizing\*\* $\mathcal{L}_{\text{ConfTS}}$. For researchers primarily focus on achieving valid conditional coverage, we recommend specialized methods such as [1,2,3,4]. Notably, [1] proposes to improve conditional coverage by minimizing pinball loss, with their results demonstrating improvements in both efficiency and conditional coverage. Their approach shows similarity to our method given the connection between the efficiency gap and pinball loss.

# M  SIMULATION

In our setup, we consider a 10-class classification problem with 200-dimensional data and implement an oracle classifier that knows the true data generation process. We control the inherent uncertainty by adding Gaussian noise to the logits, where higher noise levels represent more inherent uncertainty in the classification task. We employ APS to generate prediction sets. To ensure robustness, each experiment is repeated 100 times, and we report the average results.

The results demonstrate the relationship between task uncertainty and ConfTS's effectiveness:

Table 12: ConfTS performance analysis with synthetic data. "↓" indicates smaller values are better. **Bold** numbers are superior results.

| Noise Level | Method | Average Size ↓ | CSCV ↓ |
|---|---|---|---|
| noise_std=0 | w/o ConfTS | 1.09 | 3.45 |
| | w/ ConfTS | **0.95** | **1.65** |
| noise_std=1 | w/o ConfTS | 1.20 | **2.42** |
| | w/ ConfTS | **1.07** | 4.03 |
| noise_std=2 | w/o ConfTS | 1.36 | **3.62** |
| | w/ ConfTS | **1.27** | 4.44 |

The results show that ConfTS consistently reduces the average prediction set size across all noise settings. However, ConfTS increases the coverage gap as we add noise to the logits. This suggests that ConfTS is particularly effective in improving conditional coverage when the classification task is more deterministic.

