# OpenReview forum: "Delving into Temperature Scaling for Adaptive Conformal Prediction"
_ICLR.cc/2025/Conference — ICLR 2025 Conference Withdrawn Submission_

### Official Review · Reviewer_aEFd · 2024-10-23

**Soundness:** 2
**Presentation:** 3
**Contribution:** 2
**Rating:** 5
**Confidence:** 4

**Summary:**

This paper provides a careful study of the connection between conformal prediction and confidence calibration. They found that (i) the use of confidence calibration techniques tends to result in larger prediction sets, obtained by adaptive conformal methods; and (ii) over-confident predictions obtained by choosing a small temperature parameter results in smaller prediction sets. Building on this observation, the authors suggest tuning the temperature parameter to improve the efficiency of sets while achieving the desired coverage. Indeed, numerical experiments underscore the large gain in performance compared to baseline adaptive conformal methods.

**Strengths:**

1. I found the study on the effect of confidence calibration on conformal prediction set to be valuable.
2. The authors offer a novel cost function, used for temperature tuning, to reduce the size of the prediction sets.
3. The author's goal is to reduce the prediction set size and they show a significant gain in that performance metric.

**Weaknesses:**

My major concern is that I find it problematic to promote overly confident predictions. To me, it feels that this idea goes against the goal of uncertainty quantification. As a thought experiment, suppose you have access to the oracle classifier, in which you know the true conditional class probabilities. Suppose further that there is uncertainty in the classification, i.e., the conditional class probabilities are not close to a one-hot vector. Now, if we choose to tune the temperature to produce overconfident predictions, we may indeed obtain smaller sets, but this comes at the cost of changing the optimal conditional class probabilities, which does not make much sense. In this case, I expect that the proposed method would yield too small prediction sets that do not reflect the true uncertainty. Am I correct?

If this is the case, it would be great to discuss how the proposed method balances the trade-off between efficiency (smaller prediction sets) and accurately representing uncertainty, especially in cases where there is inherent uncertainty in the true class probabilities.

Also, I believe the authors should better clarify when one would be interested in using their approach, e.g. perhaps when one assumes that the labels are nearly a deterministic function of the images. It will be great to conduct such a synthetic experiment and analyze what is the effect of the proposed approach on conditional coverage when the oracle classifier is available.

Please don't get me wrong, it's an interesting paper that I enjoyed reading and the results are impressive. But there are crucial points that must be clarified, especially the fact that this paper advises to make overconfident predictions (unless I misunderstood it).

**Questions:**

Related to the above, I don't understand definition 4.1 and eq. 8: minimizing the distance between the quantile score \tau and all other scores will push all the scores to be very close to each other. This goes against the concept of adaptivity, where one seeks large scores for images that are harder to predict and smaller scores for images that are easier to predict.

It would be great to provide more detailed explanation of how the proposed method maintains adaptivity despite minimizing the distance between scores. This can be done, for example, by building synthetic data that inherently have easy- and hard-to-predict samples.

I don't understand why the proposed method improves conditional coverage (line 414). It would be great if the authors could explain the mechanism by which the proposed method improves conditional coverage. I suspect the authors observed this behavior when using SSCV because the partitioning of the set sizes should not be fixed: the baseline methods have larger set sizes whereas the proposed method has smaller sets so each partition contains an unbalanced number of samples across the methods. Also, it would be good to report the class-conditional coverage.

---

> ### Author Response · Authors · 2024-11-20
> **Response to Reviewer aEFd**
>
> **1. ConfTS preserves uncertainty [W1,2]:**
>
> Thank you for raising this important concern about uncertainty quantification. We would like to address this from two key aspects:
> *  Our method does not conflict with confidence calibration, as it only replaces the temperature value. During inference, one may use different temperature values according to the objective, whether for improved calibration performance or efficient prediction sets.
> * ConfTS maintains the adaptiveness property of adaptive conformal prediction. In particular, as shown in Figure 3 (a)&(b), our method produces smaller prediction sets for easy examples and larger sets for hard ones. This indicates that while ConfTS optimizes for efficiency, it still preserves the instance uncertainty where ambiguous cases receive larger prediction sets.
>
> These properties indicate that ConfTS does not compromise uncertainty quantification.
>
> **2. The performance of ConfTS when the oracle classifier is available [W3]:**
>
> Thank you for this insightful suggestion. We designed a synthetic experiment with controlled uncertainty conditions to analyze this question. In the experiment, we consider a 10-class classification problem with 200-dimensional data and implement an oracle classifier that knows the true data generation process. We control the inherent uncertainty by adding Gaussian noise to the logits, where higher noise levels represent more inherent uncertainty in the classification task. We employ APS to generate prediction sets. To ensure robustness, each experiment is repeated 100 times, and we report the average results. The results demonstrate the relationship between task uncertainty and ConfTS's effectiveness.
>
> | Noise Level | Method | Average Size | CSCV |
> |-------------|---------|--------------|--------------|
> | noise_std=0 | w/o ConfTS | 1.09 | 3.45 |
> |             | w/ ConfTS  | 0.95 | 1.65 |
> | noise_std=1 | w/o ConfTS | 1.20 | 2.42 |
> |             | w/ ConfTS  | 1.07 | 4.03 |
> | noise_std=2 | w/o ConfTS | 1.36 | 3.62 |
> |             | w/ ConfTS  | 1.27 | 4.44 |
>
> The results show that **ConfTS consistently reduces the average prediction set size** across all noise settings. However, ConfTS increases the CSCV as we add noise to the logits. This suggests that ConfTS is particularly effective in improving conditional coverage when the classification task is more deterministic. We add this experiment in Appendix M.
>
> **3. Why minimizing the efficiency gap preserves adaptivity [Q1,2]:**
>
> There might be some misunderstandings. We clarify that adaptivity, as defined in prior work [1,2,3], requires prediction sets to communicate instance-wise uncertainty such that easy examples obtain smaller sets than hard ones. Notably, adaptivity is a property required by prediction sets, instead of that for non-conformity scores. Therefore, **the distance between non-conformity scores could be unrelated to the concept of adaptivity**.
>
> In our experiments, we employ the label order (the position of the true label y in the sorted softmax probabilities π(x)) to reflect instance difficulty. This is because when a model is more confident about its prediction, the true label tends to have a higher softmax probability and thus a lower label order, and vice versa. Therefore, we can effectively examine how prediction set sizes vary with instance difficulty by partitioning samples in label order (1, 2-3, 4-6, 7-10, 11-100, 101-1000) following [2]. As demonstrated in Figure 3a and 3b, ConfTS maintains this adaptiveness property: examples with lower label order (easier samples) consistently obtain smaller prediction sets. This confirms that our method maintains adaptiveness while improving prediction efficiency.
>
> **4. The performance of ConfTS on conditional coverage [Q3]:**
>
> See general response.
>
> **References**
>
> [1] Yaniv Romano, et al. Classification with valid and adaptive coverage. NIPS 2020.
>
> [2] Anastasios Nikolas Angelopoulos, et al. Uncertainty sets for image classifiers using conformal prediction. ICLR 2021.
>
> [3] Nabeel Seedat, et al. Improving adaptive conformal prediction using self-supervised learning. ICML 2023.

---

> > ### Comment · Reviewer_aEFd · 2024-11-26
> > **Follow-up**
> >
> > > Experiments with an oracle classifier
> >
> > Thank you for this interesting experiment. It is in line with my intuition and understanding: the proposed method is useful when the task is more deterministic. I urge the authors to clarify this EARLY ON in the manuscript.
> >
> > > Regarding conditional coverage
> >
> > Thank you for your detailed response. The new analysis calrifes why by minimizing $\mathcal{L} _ {\mathrm{ConfTS}}(x,y;t)=(\tau(t)-\mathcal{S}(x,y,t))^2,$ one can improve conditional coverage. This is due $\mathrm{MSCE}\leq\mathcal{O}(\mathbb{E}[\sqrt{\mathcal{L}_{\mathrm{ConfTS}}(x,y;t)}]).$ However, notice that this optimization can push all the scores $S_i$ to have the same value, equal to $\tau$. This, in turn, would result in meaningless prediction sets, as all the scores would have the same value, regardless of $(X,Y)$. I believe temperature scaling is not flexible enough, and so the authors do not face this issue.
> >
> > Overall, the authors suggest an interesting approach to reduce the prediction set size. I am still confused about the issue with conditional coverage: the synthetic experiment shows this method is useful when the problem is more deterministic. Yet, the new conditional coverage analysis shows that by minimizing $|S-\tau|$ one can improve conditional coverage. I would love to get the author's feedback on that matter before making my final decision.

---

> ### Author Response · Authors · 2024-11-27
> **Response to Reviewer aEFd (follow-up question)**
>
> Thank you for the patient discussion. In the following, we respond to the concerns point by point.
> **1. Experiments with an oracle classifier**
> Thank you for the suggestion. In Section 5 of the revised paper, we add the statement in the analysis of conditional coverage.
>
> **2.  The effect of the proposed loss**
> There might be some misunderstanding. Firstly, we have a general assumption that the score function is not a constant function. Then, our loss function is $\mathcal{L}(x,y,t) = (\tau(t)-\mathcal{S}(x,y,t))^2$, where $\mathcal{S}(x,y,t)$ denotes the scores of examples with their **true labels** (not all labels) and $\tau$ is the $1-\alpha$ quantile of those scores. Thus, the optimization of the proposed loss pushes the scores of **ground-truth labels** (not all labels) to be close to the threshold $\tau$ for all examples. Ideally, the optimal case is that all the scores of true labels have the same value (as $\tau$), which leads to prediction sets that exactly cover the true labels (the smallest prediction set that contains the ground-truth label).
>
> **3. The limitation of the theoretical analysis**
> Thank you for raising this concern. In the presence of noise, we can decompose the ConfTS loss $|S'-\tau'|$ into two components: the clean ConfTS loss $|S-\tau|$ and the inherent noise term (the distance between noisy and clean scores). As the noise level increases, the inherent noise term becomes dominant, making the upper bound on conditional coverage less tight. Consequently, while our theoretical results hold in general, the practical benefits are more pronounced in deterministic settings where the noise term is minimal.

---

> > ### Comment · Reviewer_aEFd · 2024-12-03
> > **Reply**
> >
> > > The optimization of the proposed loss pushes the scores of ground-truth labels (not all labels) to be close to the threshold $\tau$ for all examples.
> >
> > My point is that forcing $\mathcal{L}(x,y,t) = (\tau(t)-\mathcal{S}(x,y,t))^2$ to be small can improve conditional coverage, but, in theory, the sets may become larger. This is to highlight why your restriction to optimize only the temperature is sensible; too much flexibility in this optimization process could even result in trivial sets - unless you also optimize for the set size. For example, consider an extreme case where, as a result of such an optimization process (without restriction to temp. scaling, or small set sizes), the scores become Uniform[0.90,0.91], regardless of x,y. In this case, $\mathcal{L}(x,y)$ is small, you will obtain conditional coverage, but the set size will be trivial for $(1-\alpha)*100$% of the points.
> >
> > In sum, thank you for the detailed rebuttal and active discussion. The paper suggests a sensible approach to improve the conformal set sizes in deterministic settings. I have decided to increase my score to slightly above the acceptance threshold, reflecting its borderline nature.

---

> > > ### Author Response · Authors · 2024-12-03
> > >
> > > Thank you for the discussion and raising the score. In this work, the proposed loss is designed for optimizing the parameters of scaling methods, which have limited flexibility as you mentioned. It will be an interesting direction for future works to formally analyze its effects in full training. Besides, we'd like to clarify that the main contribution of this work is revealing the effects of confidence calibration on conformal prediction, both of which are two popular frameworks of uncertainty estimation in deep learning. Then, we propose the method to show how the two frameworks can work together.
> > >
> > > Thank you again for the active discussion. We are happy that the reviewer can increase the score after the discussion (note that the rating has not been updated yet).

---

### Official Review · Reviewer_AKYr · 2024-10-29

**Soundness:** 3
**Presentation:** 2
**Contribution:** 3
**Rating:** 6
**Confidence:** 4

**Summary:**

The submission explores the limitations of conventional temperature scaling and addresses them using the conformal prediction framework. Specifically, the authors mention that temperature scaling and other post-hoc calibration methods often lead to less efficient prediction sets in adaptive conformal prediction for classification. The authors introduce conformal temperature scaling (ConfTS). The method uses optimization to minimize an “efficiency gap”, which measures the difference between the threshold and the non-conformity score of the true label. The authors claim that ConfTS significantly improves prediction set efficiency.

**Strengths:**

Based on my knowledge, this is the only work that applied conformal prediction for temperature scaling. Based on the authors’ response, I would be willing to increase the score.

The strength of the submission lies in both theoretical and empirical analysis. The theoretical analysis of the upper bound (Thm. 3.3) is interesting, although there is no indication of how to choose c_1 or \gamma in practice (there is also no empirical confirmation of this thm). The strengths of ConfTS lie in 1) compatibility for many classification models, 2) low compute, 3) does not require hyperparameter tuning, and 3) maintaining coverage guarantees while reducing the size of prediction sets. ConfTS balances prediction confidence, achieving calibration, and efficiency in multiclass settings.

**Weaknesses:**

A major weakness of this submission is that it solely concentrates on temperature scaling without motivation as to why temperature scaling is a worthwhile problem to tackle compared to other methods.
Another weakness of the submission is the limited real-world examples/case studies.
Another weakness is that some of the claims made in the main results section were not well theoretically/empirically justified (see below).
A weakness of the method lies in the assumption of homogeneity across classes - i.e., assuming that all classes require the same level of adjustment.
A weakness is that the manuscript offers no practical recommendation on transferring Thm. 3.3 into practice - c_1 and \lambda are both arbitrary values, and there is a disconnect between theory and application.

**Questions:**

Please address the main weaknesses mentioned above:
1. Why was temperature scaling chosen compared to other methods? Readers may benefit from a more comprehensive comparison with alternative calibration techniques (for example: platt scaling, vector scaling, isotonic regression). It is clear from Table 1 that temperature scaling does not perform the best out of the 4 methods. An in-depth discussion of why temperature scaling is better/worse than others and was chosen would significantly enhance the motivation of this paper.
2. When is ConfTS guaranteed to be better than other methods (like different dataset characteristics)? Because there is no guarantee that ConfTS would theoretically be better than existing methods, the submission could benefit from concrete, real-world examples showing how ConfTS would be applied in practical scenarios, including edge cases (settings with numerical instability etc.). Furthermore, there is wide variability on the performance improvements across different models and datasets. A discussion/theoretical analysis of why this could be the case would significantly improve the conviction of the proposed work.
3. What is the justification for using the same temperature scaling value for all classes? This assumption could work well in simpler cases but might not capture the variability needed for more complex or imbalanced datasets. Furthermore, the efficiency could improve as well. This was not sufficiently explored. Some empirical results/discussion about this issue would improve comprehensiveness.
4. How does one use Thm 3.3? How does one pick c_1 and \gamma ?

There are also a few questionable claims. Please comment on the following:
1. Requires no additional computational costs on temperature scaling (page 2): the optimization procedure is a computational cost. This claim may be unwarranted.
2. Improves conformal training (ConfTr): The results show that ConfTS improves ConfTR by <1 most of the time, which is extremely marginal. Please contemplate and discuss whether the average size values are significant, both statistically and in practice.
3. Improves conditional coverage: This claim is unsubstantiated because 1) there is no theoretical motivation or justification for why ConfTS produces better conditional coverage, and 2) this does not hold in all cases (see Fig 2, ViT-B-16). This would, at best, be a discussion item and not a key result.

Other minor comments:
1. t_0 and other variables need to be defined in proposition 3.1
2. Variables need defining in Thm 3.3
3. Why is the related work at the end of the manuscript?

---

> ### Author Response · Authors · 2024-11-20
> **Response to Reviewer AKYr**
>
> **1. Choice of temperature scaling [Q1]:**
>
> There might be some misunderstanding. We clarify that this work is the first to study **temperature scaling in conformal prediction**, instead of applying conformal prediction for temperature scaling. Another point we need to clarify is the results of Table 1, where we show that all the calibration methods lead to large prediction sets in adaptive conformal prediction. This observation motivates us to investigate how high-confidence predictions affect conformal prediction.
>
> Among these methods, we pick temperature scaling as it is the simplest method to adjust the confidence level with only a temperature parameter $T$. This enables us to provide a thorough analysis with theoretical and empirical results, revealing the relationship between confidence calibration and conformal prediction.
>
> In addition, we also provide new results to show the advantage of ConfTS over the variant of vector scaling (ConfVS) trained by our loss. The results below show that temperature scaling is a better choice for this task as a simple and effective solution, while ConfVS can also improve efficiency.
>
> | Model      | Vanilla | APS + ConfTS   | APS + ConfVS  |
> |------------|---------|----------------|---------------|
> | ResNet18   |  14.1   | **7.53**       | 9.20          |
> | ResNet50   |  9.06   | **4.79**       | 4.97          |
> | VGG16      |  11.7   | **6.05**       | 8.04          |
> |DensetNet121|  6.95   | **4.74**       | 6.72          |
> | Average    |  10.5   | **5.78**       | 7.23          |
>
> **2. Clarification on the performance of ConfTS [Q2]:**
>
> We clarify that ConfTS is designed as a complementary method rather than a competing alternative to existing approaches. The core contribution of our work is the observation that temperature values optimized for confidence calibration are not optimal for adaptive conformal prediction. We then propose a method to identify the optimal temperature specifically for adaptive conformal prediction. Our experimental results demonstrate that ConfTS can enhance various existing methods, including post-hoc and training approaches. We would greatly appreciate it if you could provide any suggestions for additional baseline methods or specific real-world datasets.
>
> **3. Using the same temperature for all classes [Q3]:**
>
> Thank you for this insightful suggestion about class-wise temperature scaling. In this work, we adopt the standard temperature scaling in the analysis and method design, for simplification and broad application (class-wise temperature scaling is not a common practice currently). While we acknowledge that class-specific temperature scaling could be beneficial, we employ a single temperature parameter in our method given several practical tradeoffs:
> * Optimizing class-specific temperatures would require significantly more validation data to obtain reliable estimates for each class, which is particularly challenging for classes with fewer samples in imbalanced datasets.
> * The optimization process for multiple temperature parameters would substantially increase training time, potentially limiting the method's practical applicability.
>
> We believe class-wise temperature scaling might be an interesting direction for future work, but it does not affect the contribution of this work, as demonstrated in the reply to the last concern.
>
> **4. Interpretation of Theorem 3.3 [Q4]:**
>
> Thank you for pointing out this important question. Theorem 3.3 serves primarily as theoretical motivation for our method. In particular, it demonstrates that high-confidence predictions produced by a small temperature could lead to efficient prediction sets on expectation. The theorem includes two constants, $c_1$ and $\lambda$, which are determined by the distribution of non-conformity scores $S$ under the assumptions detailed in Appendix F.3. Similar constants also appear in Theorem 14 of previous work [1]. While these constants are important for the theoretical analysis, our practical method does not require their explicit calculation. Instead, the theorem provides the key insight that guides our approach: optimizing temperature can enhance the efficiency of adaptive conformal prediction. This theoretical foundation is validated by our empirical results in Figure 1.
>
> **5. Clarification on computational cost [Q5]:**
>
> Thank you for pointing out the potential confusion. For classification, temperature scaling has been an essential part in the workflow of deep learning, as a common practice. Therefore, our method does not require additional computational cost compared to the standard temperature scaling, which also requires optimization [2]. Our method only modifies the training objective of the optimization, rather than introducing a new procedure.

---

> ### Author Response · Authors · 2024-11-20
> **Response to Reviewer AKYr**
>
> **6. The statistical and practical significance of the ConfTS improvement over ConfTr [Q6]:**
>
> Thank you for this great suggestion. Let us address both statistical and practical significance.
>
> **Statistical significance:** In Table 3, we provide the results of two-sample t-test at a 0.1 confidence level. The p-values for our comparisons are presented in the table below. The small p-values (<0.1) across all tests indicate that these improvements are statistically significant.
>
> | Model    | Method | $\alpha=0.1$ p-value | $\alpha=0.05$ p-value |
> |----------|--------|---------------|----------------|
> | ResNet18 | APS    | 1.34e-07     | 4.04e-10      |
> |          | RAPS   | 2.60e-04     | 7.71e-03      |
> | ResNet50 | APS    | 7.11e-04     | 3.11e-07      |
> |          | RAPS   | 7.71e-03     | 4.58e-03      |
>
> **Practical significance:** As established in our core contribution, we demonstrate that temperature values optimized for confidence calibration are suboptimal for adaptive conformal prediction, and we propose a method to identify the optimal temperature specifically for this task. The consistent improvements observed in the experiment of ConfTr further validate our conclusion in the training method. Moreover, as demonstrated in previous work [3], the reduction in the prediction set size has practical significance, as smaller prediction sets are more informative to enable accurate human decision making.
>
> **7. The performance of ConfTS on conditional coverage [Q7]:**
>
> See general response.
>
> **8. Minor comments:**
>
> We fix these issues accordingly. It is a common practice to put the related work at the end of the manuscript so that readers can quickly touch the definition of problem setting.
>
> **References**
>
> [1] Sadinle M, et al. Least ambiguous set-valued classifiers with bounded error levels. JASA 2019.
>
> [2] Guo C, et al. On calibration of modern neural networks. ICML 2017.
>
> [3] Jesse C, et al. Conformal Prediction Sets Improve Human Decision Making. ICML 2024.

---

> > ### Comment · Reviewer_AKYr · 2024-11-25
> >
> > Thank you for your response, clarifications, and revised submission. After careful consideration, I have decided to raise your score.

---

> > > ### Author Response · Authors · 2024-11-27
> > >
> > > Thank you for your valuable feedback that helped improve our manuscript.

---

### Official Review · Reviewer_k1qr · 2024-11-01

**Soundness:** 3
**Presentation:** 3
**Contribution:** 3
**Rating:** 6
**Confidence:** 3

**Summary:**

The paper proposes a variant of temperature scaling ConTS to enhance adaptive conformal prediction by minimizing the gap between the threshold and the non-conformity score of the ground truth for a held-out validation dataset. Experiments in the paper show that ConfTS can effectively reduce prediction set sizes of APS and RAPS while maintaining marginal coverage. Moreover, it can provide prediction sets with better conditional coverage.

**Strengths:**

The paper introduces an effective improvement over existing temperature scaling, aiming for efficiency in conformal prediction by minimizing the gap between the threshold and the non-conformity score. A solid theoretical foundation for ConfTS is also provided.

The paper is well-written and clearly structured.

**Weaknesses:**

1. Like the author mentioned, ConfTS does not directly address scenarios where distribution shifts occur, such as changes in data distribution over time.

2. While effective in multi-class settings, ConfTS’s performance in binary classification or non-image-based domains is less explored.

**Questions:**

1. How does ConfTS perform in non-image prediction tasks?

2. Would there be a way to adapt ConfTS to handle distribution shifts effectively, perhaps by dynamic temperature adjustments?

---

> ### Author Response · Authors · 2024-11-20
> **Response to Reviewer k1qr**
>
> **1. Results of non-image prediction tasks [Q1]:**
>
> Thank you for this important question. We evaluated ConfTS on text classification using the AG News dataset [1] with a pre-trained BERT model [2] fine-tuned for one epoch. We use APS and RAPS ($\lambda=0.001$) to generate prediction sets, and the results demonstrate consistent improvements, which demonstrates the effectiveness of ConfTS beyond image tasks.
>
> | Method | Alpha | vanilla  |  + ConfTS |
> |--------|-------|----------|-----------|
> | APS    | 0.1   | 1.04     | **1.03**      |
> |        | 0.05  | 1.15     | **1.13**      |
> |        | 0.01  | 1.53     | **1.43**      |
> | RAPS   | 0.1   | 1.03     | **1.00**      |
> |        | 0.05  | 1.14     | **1.09**      |
> |        | 0.01  | 1.53     | **1.49**      |
>
>
>
> **2. Handling distribution shifts with online conformal prediction [Q2]:**
>
> Thank you for this insightful question about handling distribution shifts. One promising approach would be to integrate ConfTS with online conformal prediction [3], where the temperature parameter could be dynamically adjusted as new data arrives. This could potentially help maintain optimal prediction set sizes even as the underlying distribution evolves. We believe this represents an interesting direction for future research.
>
>
> ## References
> [1] Xiang Zhang, et al. Character-level Convolutional Networks for Text Classification. NIPS 2015.
>
> [2] Jacob Devlin, et al. BERT: Pre-training of Deep Bidirectional Transformers for Language Understanding. NAACL-HLT 2019.
>
> [3] Isaac Gibbs, et al. Adaptive Conformal Inference Under Distribution Shift. NIPS 2021.

---

> > ### Comment · Reviewer_k1qr · 2024-12-02
> >
> > Thank you for addressing most of the concerns. I will maintain score for this paper.

---

### Official Review · Reviewer_VZ7E · 2024-11-03

**Soundness:** 2
**Presentation:** 2
**Contribution:** 1
**Rating:** 3
**Confidence:** 4

**Summary:**

The paper is well-written and the author has conducted many experiments of ConfTS in CIFAR100 and ImageNet with various architectures. The post-processing  of conformal prediction is also interesting.

**Strengths:**

The paper is well-written and the author has conducted many experiments of ConfTS in CIFAR100 and ImageNet with various architectures. The post-processing  of conformal prediction is also interesting.

**Weaknesses:**

. In lines 25-26, the paper claims a nearly 50% reduction in the average size of APS and RAPS on ImageNet at an error rate of $\alpha$ = 0.1. This assertion requires clarification regarding the specific model architectures involved. Does this reduction apply universally across all models, or only to the specific architectures tested in the paper, such as ResNet18 and ViT-B-16?

2. **The experiments seem to be conducted improperly**. In the RAPS paper [1], Appendix E details the method for optimizing set size by choosing $k_{reg}$ and $\lambda$. However, in your paper (Page 16, line 859), these parameters are fixed $k_{reg} = 1$ and $\lambda = 0.001$, potentially inflating  the average set size of RAPS.

3. The experimental setup creates an **unfair baseline**. For instance, in the RAPS paper, the ResNet18's set size at a 0.1 error rate is 4.43 (Table 1 in [1]), while your results show inefficiency of 9.605 (base) and 5.003 (after ConfTS) at error rate 0.1. This indicates that the tuned RAPS results after ConfTS are worse than the baseline reported in [1]. Similar issues appear for other models like ResNet50 and ResNet101, where inefficiencies post-ConfTS remain higher than those reported in RAPS. Please refer to Table 1 in [1] for a comparison.

4. In line 291, you optimize the optimization of $T$ via loss in equation (8). Given that $T$ is a scalar, a grid search on a held-out validation set might be a more efficient strategy. This approach is particularly feasible for small datasets.

5. The paper compares the baseline with ConfTr [5]. Can ConfTr be employed in tuning $T$?

6.  It's essential to consider the rationale of excessive tuning of $T$. APS is designed to assure conditional coverage with an oracle classifier [2] (Section 1.1). Excessive tuning might skew the non-conformity scores of APS and RAPS towards  Least Ambiguous Set-valued Classifier (LAC) [3], potentially compromising the original intention of APS and RAPS.

7. Does tuning $T$ affect the class-stratified coverage gap (CSCG) [6], another vital adaptiveness metric alongside SSCV? Lower values of $T$ might reduce CSCG for RAPS.

8. Have you evaluated the SSCV of ConfTS for RAPS? It’s crucial to determine whether it enhances adaptiveness for more strong baseline.

9. There appears to be an error in the second column of Table 4, where APS and RAPS are listed both in the row and the column. Please correct this for clarity.

References:

[1] Anastasios Nikolas Angelopoulos, Stephen Bates, Michael I. Jordan, and Jitendra Malik. Uncertainty sets for image classifiers using conformal prediction. In 9th International Conference on Learning Representations, ICLR 2021,

[2]Yaniv Romano, Matteo Sesia, and Emmanuel Candes. Classification with valid and adaptive coverage. In Advances in Neural Information Processing Systems, 33:3581–3591, 2020.

[3] Mauricio Sadinle, Jing Lei, and Larry Wasserman. Least ambiguous set-valued classifiers with bounded error levels. Journal of the American Statistical Association, 114(525):223–234, 2019.

[5]David Stutz, Krishnamurthy Dvijotham, Ali Taylan Cemgil, and Arnaud Doucet. Learning optimal conformal classifiers. In The Tenth International Conference on Learning Representations, ICLR 2022.

[6] Tiffany Ding, Anastasios Angelopoulos, Stephen Bates, Michael Jordan, and Ryan J Tibshirani. Classconditional conformal prediction with many classes. Advances in Neural Information Processing Systems, 36, 2024

**Questions:**

See the above.

---

> ### Author Response · Authors · 2024-11-20
> **Response to Reviewer VZ7E**
>
> **1. The results of ConfTS on ImageNet [W1]:**
>
> We clarify that the 'nearly 50% reduction' refers to the average improvement across all six model architectures tested as shown in Table 2. In particular, we conduct experiments on various model architectures: ResNet18, ResNet50, ResNet101, DenseNet121, VGG16, and ViT-B-16. To avoid any misunderstandings, we revised the sentence (lines 25-26) as: 'When averaged across six different architectures, ConfTS reduces the size of APS and RAPS on ImageNet by nearly 50% at an error rate of α = 0.1.'
>
> **2. The results on RAPS [W2,3]:**
>
> Thank you for raising this important concern about experimental fairness. In our experiment, we set $k_{reg}=1$ because this setting consistently achieves smaller average size in Table 3 of their paper [1]. In addition, we use the same setting to ensure a fair comparison when comparing the average size of RAPS with and without ConfTS. This enables us to analyze the impact of our method in consistent conditions of baselines. As for $\lambda$, we provide extensive results in Appendix I to demonstrate the consistent performance of ConfTS across different $\lambda\in$ {0.002, 0.004, 0.006, 0.01, 0.015, 0.02}.
>
> As you suggested, we also conduct experiments to compare the **best** performance between RAPS with and without ConfTS with **tuned** $\lambda$. The results show that ConfTS still achieves improvements for RAPS in average set size:
>
> | Model      | RAPS [1] | RAPS + ConfTS (Appendix I) |
> |------------|----------------------|---------------------|
> | ResNet18   | 4.43                | **4.28**               |
> | ResNet50   | 2.57                | **2.32**               |
> | ResNet101  | 2.25                | **2.10**               |
> | VGG16      | 3.54                | **3.34**               |
> | Average    | 3.20                | **3.01**               |
>
>
> **3. Why we adopt the optimization for searching T [W4]:**
>
> In temperature scaling, using optimization to search the temperature value has been a common practice in deep learning [2]. While grid search is indeed a viable approach for scalar optimization, we choose gradient-based optimization for several reasons:
> * Grid search can only explore a discrete domain, like $t\in[1.0,0.9,0.8,0.7,0.6]$, which makes it challenging to search for the optimal temperature;
> * Grid search can be labor-intensive and requires tuning for each dataset or task;
> * Gradient-based optimization can be applied to other non-scalar methods, like vector scaling [2];
>
> Therefore, our method can be treated as an automatic and adaptive tuning method, which requires less manual intervention than Grid search and can be extended to other complex methods of calibration for broad application.
>
> **4. Using ConfTr to tune temperature value [W5]:**
>
> Thank you for this interesting suggestion. However, ConfTr is not suitable for our setting because it would drive the temperature parameter toward extremely small values. As shown in Figure 1 \(c), such small temperature values cause the prediction sets to become overly conservative.
>
> | Model      | APS + ConfTS   | APS + ConfTr  |
> |------------|----------------|---------------|
> | ResNet18   | **7.53**           | 979.3         |
> | ResNet50   | **5.00**           | 985.4         |
> | ResNet101  | **4.79**           | 983.2         |
>
> **5. The performance of ConfTS on conditional coverage [W6,7,8]:**
>
> See general response.
>
> **6. Typo in Table 6 [W9]:**
>
> Thank you for pointing out this potential source of confusion. To clarify: the rows indicate the methods used to generate the prediction sets, while the columns indicate the scoring functions used in the ConfTS optimization process. We have revised the caption of Table 4.
>
> **References**
>
> [1] Angelopoulos A, et al. Uncertainty sets for image classifiers using conformal prediction. ICLR 2021
>
> [2] Guo C, et al. On calibration of modern neural networks. ICML 2017.

---

> > ### Comment · Reviewer_VZ7E · 2024-11-28
> >
> > Thanks for your response. However, I decided to maintain my score for the following reasons:
> > 1. **Inconsistent Baselines**: As you have acknowledged, the experimental configurations are different from RAPS [1], and the reported average set size in the main paper appears inflated. Thus, it is difficult to draw convincing conclusions based on such a biased comparison.
> > 2. **Omission of Grid Search Results**: Grid search results are missing. As suggested in [2], binary search is a practical approach for hyperparameter tuning. I strongly recommend including experiments with proper grid search settings for a more rigorous evaluation.
> > 3. **Lack of Adaptiveness**: ConfTS does not extend to RAPS for SSCV and CSCV, as noted in my earlier comment [W8]. If this impacts the adaptiveness, why not consider the Least Ambiguous Set-valued Classifier (LAC)? While your method may improve LAC's average set size, this analysis is absent from the current paper.
> >
> > In addition, I performed a binary search over the temperature scaling within the range [0.1, 1] using the same setting in RAPS by adding a few additional lines of grid search code in [3]. Below is the average set size for RAPS and RAPS + Grid Search. Clearly, RAPS + Grid Search achieves better set size than what you reported for ConfTS.
> > | Model     | RAPS [1] | RAPS + ConfTS (Appendix I) | RAPS + Grid Search |
> > |-----------|----------|----------------------------|--------------------|
> > | ResNet50  | 2.57     | 2.32                       | 2.22               |
> > Although your method might work well for vector scaling, you have not combined the vector scaling with a correct setting of RAPS. Therefore, I would like to hear your feedback on these matters and then make the final decision.
> >
> > [1] Anastasios Nikolas Angelopoulos, Stephen Bates, Michael I. Jordan, and Jitendra Malik. Uncertainty sets for image classifiers using conformal prediction, ICLR 2021,
> > [2] Radford, Alec, et al. "Learning transferable visual models from natural language supervision." International conference on machine learning. PMLR, 2021.
> > [3] RAPS official code: https://github.com/aangelopoulos/conformal_classification

---

### Author Response · Authors · 2024-11-20
**General Response**

We sincerely thank all reviewers for their time, insightful suggestions, and valuable comments. We are encouraged that the reviewers recognize that our work is **clear and well-structured** [VZ7E, k1qr]. We also encouraged that the reviewers find our method **novel and interesting** [VZ7E, AKYr, aEFd] with various **practical** benefits [AKYr], demonstrating **significant** improvements in performance [aEFd] on **comprehensive** benchmarks [VZ7E, AKYr]. Furthermore, the reviewers highlight that our conclusions are **valuable** [aEFd], and supported by **solid** theoretical foundation [k1qr, AKYr].

We provide point-by-point responses to all reviewers’ comments and concerns. The reviews allow us to strengthen our manuscript and the changes are summarized below:
* Revised the claims about efficiency improvements (lines 25-26).
* Clarified variable definitions in Section 3.3.
* Updated Table 4's caption for better clarity.
* Added detailed justification for the choice of temperature scaling in Appendix E.
* Expanded discussion of conditional coverage performance in Appendix L.
* Included synthetic data experiments in Appendix M.

For clarity, we highlight the revised part of the manuscript in blue color.

Here, we address the reviewers' main concern regarding the performance of ConfTS on conditional coverage.

**The performance of ConfTS on conditional coverage:**

We can formally show why ConfTS enhances conditional coverage for APS. Following [1], we use Mean Squared Conditional Error (MSCE) as a measure of conditional coverage performance:
$$\mathrm{MSCE}=\mathbb{E} _ {x\sim\mathcal{X}}[\left\\{\mathrm{Coverage(\mathcal{C}(X)|X=x)}-(1-\alpha)\right\\}^2]$$
In particular, it quantifies how prediction sets deviate from the ideal conditional coverage:
$$\mathbb{P}\\{Y\in\mathcal{C}(X)|X=x\\}=1-\alpha$$
As shown in [1], MSCE is a valid measure of conditional coverage performance. Consider the pinball loss [3]:
\begin{align}
l_{1-\alpha}(\tau,s)
=\alpha(\tau-s)\mathbf{1}\\{\tau\geq s\\}+(1-\alpha)(s-\tau)\mathbf{1}\\{\tau\leq s\\}.
\end{align}
where $\mathbf{1}\\{\cdot\\}$ is the indicator function. Through a simple corollary of Proposition 3.5 in [1], we know that the MSCE is upper bounded by the expectation of pinball loss $\mathcal{O}(\mathbb{E}[l_{1-\alpha}(\tau,S)])$, where the expectation is taken with respect to the randomization in $S$. Furthermore, by definition, the pinball loss is upper bounded by the efficiency gap $\mathcal{O}(|\tau-S|)$ (by the definition). Recall that the ConfTS loss is defined as
$$\mathcal{L} _ {\mathrm{ConfTS}}(x,y;t)=(\tau(t)-\mathcal{S}(x,y,t))^2,$$
which leads to our key conclusion:
$$\mathrm{MSCE}\leq\mathcal{O}(\mathbb{E}[\sqrt{\mathcal{L}_{\mathrm{ConfTS}}(x,y;t)}])$$
Thus, we conclude that by minimizing $|\tau-S|$ and consequently reducing MSCE, ConfTS improves conditional coverage. We add this discussion to Appendix L, and the rigorous proof can be found in Appendix G.4.

**Important Note:** Though ConfTS demonstrates enhanced conditional coverage, we emphasize that this is an auxiliary benefit rather than its core design purpose and we acknowledge that this improvement does not extend to RAPS in terms of SSCV and CSCV. This is because **temperature tuning alone provides limited capacity for minimizing** $\mathcal{L}_{\mathrm{ConfTS}}$. For researchers primarily focus on achieving valid conditional coverage, we recommend specialized methods such as [1,2,3,4]. Notably, [1] proposes to improve conditional coverage by minimizing pinball loss, with their results demonstrating improvements in both efficiency and conditional coverage. Their approach shows similarity to our method given the connection between the efficiency gap and pinball loss.

## References
[1] Shayan Kiyani, et al. Conformal Prediction with Learned Features. ICML 2024.

[2] Jung C, et al. Batch multivalid conformal prediction. ICLR 2023.

[3] Ding T, et al. Class-conditional conformal prediction with many classes. NIPS 2024.

[4] Guan L. Localized conformal prediction: A generalized inference framework for conformal prediction. Biometrika, 2023.

---

### Note · Authors · 2024-12-04

I have read and agree with the venue's withdrawal policy on behalf of myself and my co-authors.